# Decoding phase separation of prion-like domains through data-driven scaling laws

M Julia Maristany[1], Anne Aguirre Gonzalez[2], Jorge R Espinosa[3], Jan Huertas[2], Rosana Collepardo-Guevara[2,4], Jerelle A Joseph[5,6]*

[1]Department of Physics, University of Cambridge, Cambridge, United Kingdom; [2]Yusuf Hamied Department of Chemistry, University of Cambridge, Cambridge, United Kingdom; [3]Department of Physical Chemistry, Universidad Complutense de Madrid, Madrid, Spain; [4]Department of Genetics, University of Cambridge, Cambridge, United Kingdom; [5]Department of Chemical and Biological Engineering, Princeton University, Princeton, United States; [6]Omenn–Darling Bioengineering Institute, Princeton University, Princeton, United States

## eLife Assessment

The authors performed extensive coarse-grained molecular dynamics simulations of 140 different prion-like domain variants to interrogate how specific amino acid substitutions determine the driving forces for phase separation. The analyses are **solid**, and the derived predictive scaling laws can aid in identifying potential phase-separating regions in uncharacterized proteins. Overall, this is a **valuable** contribution to the field of biomolecular condensates. It exemplifies how data-driven methodologies can uncover new insights into complex biological phenomena.

*For correspondence:
jerellejoseph@princeton.edu

**Abstract** Proteins containing prion-like low complexity domains (PLDs) are common drivers of the formation of biomolecular condensates and are prone to misregulation due to amino acid mutations. Here, we exploit the accuracy of our residue-resolution coarse-grained model, Mpipi, to quantify the impact of amino acid mutations on the stability of 140 PLD mutants from six proteins (hnRNPA1, TDP43, FUS, EWSR1, RBM14, and TIA1). Our simulations reveal the existence of scaling laws that quantify the range of change in the critical solution temperature of PLDs as a function of the number and type of amino acid sequence mutations. These rules are consistent with the physicochemical properties of the mutations and extend across the entire family tested, suggesting that scaling laws can be used as tools to predict changes in the stability of PLD condensates. Our work offers a quantitative lens into how the emergent behavior of PLD solutions vary in response to physicochemical changes of single PLD molecules.

## Introduction

Biomolecular condensates are highly multicomponent systems with remarkable specificity: they exhibit non-random molecular compositions and, concomitantly, contain distinct physicochemical environments defined by the collective behavior of their components (***Banani et al., 2017***; ***Li et al., 2012***). Phase separation has emerged as a leading mechanism to account for the formation of biomolecular condensates. In this thermodynamic process, segregative (density-driven) and associative (connectivity-driven) transitions are coupled to produce protein- and nucleic acid-rich phases suspended in the cellular cytoplasm or nucleoplasm (***Mittag and Pappu, 2022***). Within this framework, intrinsically disordered proteins/regions (IDP/IDRs) have emerged as strong contributors to the multivalency and phase separation capacity of many naturally occurring proteins (***Shin and Brangwynne, 2017***; ***Banani***

**eLife digest** Our cells contain tens of thousands of different proteins that carry out essential activities to keep us alive. Some of these proteins have long, flexible regions where many of the same amino acid building blocks are repeated in the sequence. These 'prion-like low-complexity domains' (known as PLDs for short) aid in organising cellular processes by helping to form concentrated droplets of proteins within cells, known as biomolecular condensates.

Mutations in PLDs can make these condensates unstable, potentially leading to harmful protein clumps, or aggregates, that are linked to neurodegenerative diseases. However, the exact impact of specific PLD mutations on health and disease remains an open question.

To investigate this, Maristany et al. used a highly accurate computational model to simulate how 140 different PLD mutations from six proteins affected condensate stability. This analysis showed that specific types of mutations, especially of amino acids with certain properties, have predictable effects on PLD condensate behaviour. The changes in stability followed consistent scaling laws across all six proteins tested, and the data and framework generated from this simulation could help to predict condensate formation in other proteins.

The findings of Maristany et al. help to explain how PLDs regulate biomolecular condensates and how their dysregulation might lead to disease. The predictive scaling laws could one day help researchers to design therapies that restore biomolecular condensate stability in neurodegenerative disorders. However, further research is needed to test whether these predictive models apply in more complex living cells and animals.

*et al., 2017*; *Gomes and Shorter, 2019*; *Nott et al., 2015*; *Smith et al., 2016*; *Wang et al., 2018*; *Martin et al., 2020*; *Martin and Holehouse, 2020*; *Posey et al., 2018*; *Elbaum-Garfinkle et al., 2015*). Moreover, IDRs characterized by amino acid sequences of low complexity and enriched in aromatic residues, such as the PLDs of RNA–binding proteins (*Franzmann et al., 2018*; *Maharana et al., 2018*; *Wang et al., 2018*; *Gotor et al., 2020*; *Molliex et al., 2015*), are prone to form condensates that behave like viscous fluids. Furthermore, the PLDs of various naturally occurring proteins, such as Fused in Sarcoma (FUS), Trans-activation response DNA-binding protein 43 (TDP-43), and the heterogeneous nuclear ribonucleoprotein A1 (hnRNPA1) have been reported to transition from liquid-like condensates into dynamically arrested solids and glasses over time (*Jawerth et al., 2020*; *Alshareedah et al., 2023*). Such arrested states have been postulated as precursors of pathological protein aggregates and amyloids (*Linsenmeier et al., 2022*). Understanding the molecular factors that regulate the ability of PLDs to phase separate is of crucial importance, as these are conjectured to also facilitate their aggregation (*Sprunger and Jackrel, 2021*).

Biomolecular condensates are networked fluids sustained by multivalent macromolecules that interconnect with more than two partners simultaneously (*Li et al., 2012*; *Banani et al., 2016*; *Choi et al., 2020a*; *Choi et al., 2020b*; *Banani et al., 2017*; *Harmon et al., 2017*). Thus, the factors that increase the molecular valency, give rise to more densely connected liquid networks, and more stable condensates (*Banani et al., 2017*; *Choi et al., 2020a*; *Iserman et al., 2020*; *Adame-Arana et al., 2020*; *Munder et al., 2016*; *Cinar et al., 2019*). For PLDs, multivalency has been shown to be strongly promoted by the presence of aromatic residues, which act as 'stickers' that enable associative $\pi$–$\pi$ interactions, and also cation–$\pi$ contacts with arginine and lysine (*Fisher and Elbaum-Garfinkle, 2020*; *Martin et al., 2020*; *Krainer et al., 2021*; *Wang et al., 2018*; *Joseph et al., 2021*; *Bremer et al., 2022*; *Brady et al., 2017*; *Vernon et al., 2018*; *Fossat et al., 2021*). Indeed, the range of the coexistence of PLD condensates has been shown to change most significantly in response to mutations involving aromatic residues (*Martin and Mittag, 2018*; *Martin et al., 2020*). Quantitative experimental phase diagrams for a set of hnRNPA1-IDR variants in combination with mean-field simulations demonstrated that tyrosine drives stronger associative interactions than phenylalanine, arginine's effects are context-specific, and lysine destabilizes PLD–PLD interactions (*Bremer et al., 2022*). More importantly, such a study demonstrates that, at least for hnRNPA1, there exist quantitative rules that explain the changes in thermodynamic parameters of condensates in response to precise amino-acid sequence mutations.

In this work, we investigate whether there are scaling laws that can predict how the critical parameters of a broader set of PLDs change in response to amino acid sequence mutations. Scaling laws

are mathematical relationships that describe how the properties of a system change for a given variable, such as its size or composition (*Ball, 1991*). In the context of polymer-like systems such as IDPs, scaling laws have offered invaluable insight into vastly different aspects of their physical behavior: for example, they were instrumental in determining that the relationship between viscosity and radius of gyration in polymer solutions is described by a power law between the two quantities *Dunstan, 2019*; demonstrating that aging scaling during polymer collapse is a universal property in homopolymers *Majumder et al., 2020*; determining that in single polymer systems, their end-to-end distance is directly proportional to their length *Rubinstein and Colby, 2003*; showing that foldable sequences with more compact unfolded states are recent products of protein evolution (*Hofmann et al., 2012*). In the context of protein condensates, we hypothesized that scaling laws, if found, may allow us to quantify how condensates respond to different intrinsic perturbations—such as amino acid mutations or presence/absence of different post-translational modifications, changes in protein length, increased/decreased flexibility—or external stimuli—such as changes to the pH of the medium or the ionic environment. Thus, we focused on determining if such laws exist for PLDs by exploring six different RNA-binding proteins (i.e. hnRNPA1, TDP43, FUS, EWSR1, RBM14, and TIA1; *Figure 1A and B*), linked to the etiology of neurodegenerative disorders through pathology and genetics (*Clarke et al., 2021*; *Kim et al., 2013*; *Wang et al., 2008*; *Jo et al., 2020*; *Ash et al., 2021*; *Apicco et al., 2018*; *Deng et al., 2014*; *Lagier-Tourenne and Cleveland, 2009*; *Couthouis et al., 2012*; *Lee et al., 2019*; *Li et al., 2013*).

Leveraging the accuracy and efficiency of a modern sequence-dependent coarse-grained model (*Joseph et al., 2021*), we quantify the phase diagrams (temperature-vs-density plane) of a large set of mutants spanning all six PLDs mentioned above. We performed Direct Coexistence molecular dynamics (MD) simulations (*Figure 1C*), exploring simultaneously the protein-rich and protein-depleted phases of the condensate for each mutant, in the temperature–concentration phase space. Our set of simulations encompass 140 PLD mutants, each sampled at 8–12 different temperatures. Accounting for independent replicas to ensure proper sampling and convergence, the dataset presented here comprises the results of approximately 2000 independent Direct Coexistence simulations (*Figure 1D*, *Appendix 1—figure 6*), totaling 1.2 milliseconds of accumulated sampling.

Our work puts forward a data-driven scaling-law framework for the study of PLD phase behavior. We reveal that the changes in the critical solution temperatures of PLDs, as a function of the number and type of amino acid sequence mutations (e.g. aromatic, positively charged, etc), can indeed be quantified by scaling laws, which are conserved across the family of PLDs we investigate. The addition of aromatic residues yields a positive linear correlation between mutation fraction and critical temperature changes, consistent with the dominant role of aromatic residues in PLD phase separation. We find that protein length dictates scaling effects when aromatic residues are mutated into uncharged, non-aromatic amino acids. We also quantify the effects of arginine mutations on condensate stability, where removing arginines decreases the critical temperature of the condensate—revealing an inverse relationship between the fraction of arginine deletions and critical temperatures. Finally, polar amino acids emerge as subtle modulators of the phase behavior of PLDs. By examining the impact of different amino acid substitutions on the critical solution temperature, our results provide insights into the underlying molecular mechanisms of protein condensation. The scaling laws presented in this study serve two purposes: first, to verify the generality of the molecular grammar that governs the phase behavior across various PLD proteins, and second, to quantify the extent to which the critical solution temperature of a PLD protein solution will change in response to specific amino acid sequence changes. We anticipate that scaling laws for other families of proteins with different compositional biases (e.g. transcription factors, nucleoporins, or molecular chaperones) may exist, but could be distinct from those we report here for PLDs. Overall, our work suggests that scaling laws represent useful predictors of the phase behavior of protein families.

## Results

### Generating and analyzing an extensive dataset of critical solution temperatures for prion-like domain variants

The phase behavior of PLDs and their modulation with amino acid sequence mutations have been extensively investigated with a variety of experimental techniques and with theory and simulation

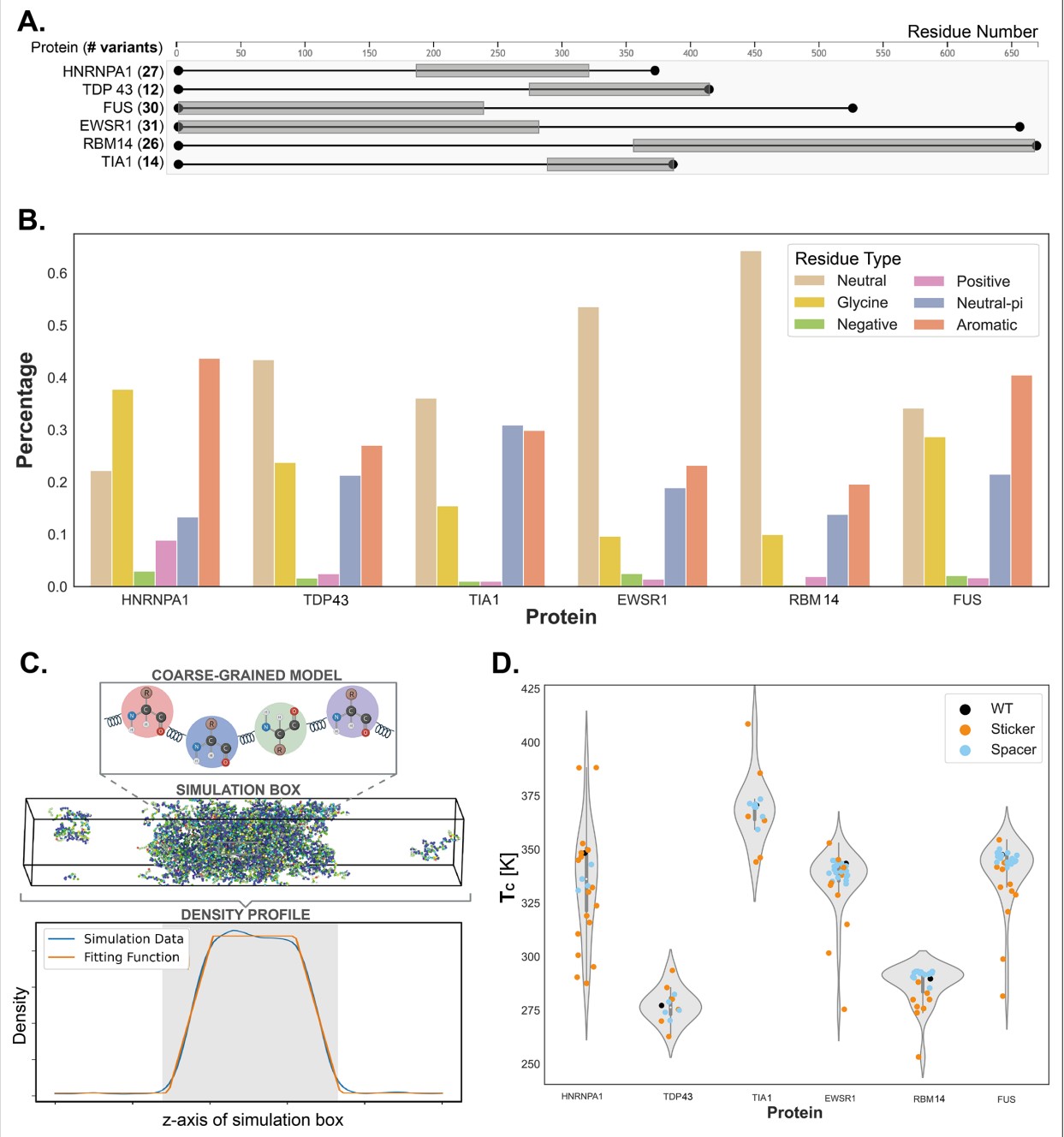

**Figure 1.** Generation and representation of dataset for probing scaling laws of prion-like low complexity domain (PLD) phase behavior.
(**A**) Representation of the prion-like domains (gray-shaded regions) of each considered protein, alongside the associated number of variants simulated, totaling a set of 140 sequences. (**B**) Composition of the wild-type sequence of the PLDs simulated, in terms of the percentage of neutral amino acids (Ala, Cys, Ile, Leu, Met, Pro, Ser, Thr, and Val; no net charge at pH 7 and no π electrons in the side chain), glycine (Gly), negative (Asp, Glu), positive (Lys, His, Arg), neutral–π (Asn and Gln; no net charge at pH 7 with π electrons in the side chain), and aromatic (Phe, Trp, Tyr) residues. (**C**) Top panel: Representation of the simulation model: each amino acid of the protein is represented by a single bead, bonded interactions are modeled with springs, and non-bounded interactions are modeled with a combination of the Wang–Frenkel potential and a Coulomb term with Debye–Hückel screening. Middle panel: The Direct Coexistence simulation method, in the slab geometry, is employed, where both the protein-rich and protein-depleted phases are simulated simultaneously. Bottom panel: Finally, to obtain the data point in the concentration–temperature phase space, at a given temperature the density profile is computed, and the concentrations of the rich and depleted phases are obtained via a suitable fitting (see Methods). (**D**) Representation of the entire computational dataset. Orange data points represent variants where charged or aromatic residues were mutated (i.e. stickers), while cyan data points represent all other types of mutations studied (i.e. spacers).

methods for systems including the hnNRPA1 PLD (*Bremer et al., 2022*; *Farag et al., 2023*), the FUS PLD (*Reber et al., 2021*; *Feng et al., 2019*; *Wang et al., 2018*; *Welsh et al., 2022*; *Luo et al., 2018*; *Murray et al., 2017*; *Qamar et al., 2018*; *Couthouis et al., 2012*; *Deng et al., 2014*; *Murakami et al., 2015*; *Patel et al., 2015*; *Murthy et al., 2019*), and the TDP-43 PLD (*Lagier-Tourenne and Cleveland, 2009*; *Zbinden et al., 2020*; *Jo et al., 2020*; *Wang et al., 2008*; *Guenther et al., 2018*), among many others. Here, we perform residue-resolution coarse-grained MD simulations (see *Figure 1C*, with further details in the Methods section) with our Mpipi model (*Joseph et al., 2021*) to quantitatively compare the critical parameters of a total of 140 mutants of the PLDs of TDP-43, FUS, EWSR1, hnRNPA1, RBM14, and TIA1 (*Figure 1A*) under consistent solution conditions. From our simulations, we compute temperature-vs-density phase diagrams and estimate the critical solution temperature for each system. The PLDs of the proteins we investigate have similar composition biases—i.e., are highly rich in neutral, polar, and aromatic residues—and present less than 5% of total charged amino acids in their sequence (*Figure 1B*). Thus, our simulation results allow us to assess if there exists a code—i.e., a simple set of scaling laws—that can predict how the critical parameters of the family of PLDs will scale in response to amino acid sequence mutations.

Our residue-resolution model Mpipi (see Methods) is well-suited to investigate the phase behavior of PLDs because it achieves the correct balance of the dominant π–π and cation–π interactions, which drives the phase behavior of PLDs. Indeed, the Mpipi model (*Joseph et al., 2021*) accurately recapitulates the quantitative experimental phase diagrams of hnRNPA1 variants (*Bremer et al., 2022*). It is worth noting that the Mpipi model was not parameterized based on the experimental dataset of Bremer and colleagues (*Bremer et al., 2022*), but rather the model parameters were developed independently by combining atomistic umbrella sampling MD simulations of amino acid pairs with the bioinformatics data of Vernon and colleagues for π-driven contacts (*Vernon et al., 2018*). Using Flory–Huggins–Staverman theory, critical temperatures for phase separation were extracted from the experimental data of *Bremer et al., 2022*, which were then used to validate the Mpipi model (*Joseph et al., 2021*).

The amino acid sequences of all the proteins we investigated (see Appendix 1) were designed based on five main considerations. (1) The set of mutants should serve to probe the effects of both the strong modulators of PLD phase separation—namely, aromatic and arginine interactions—and the subtle modulators—that is, polar and neutral amino acids, previously identified experimentally (*Bremer et al., 2022*; *Martin et al., 2020*; *Wang et al., 2018*). (2) To ensure that the observed changes in the critical solution temperatures can be attributed unequivocally to specific amino-acid-type changes, we mutated only one amino acid type at a time. However, multiple mutations of the same amino acid type were allowed. (3) Mutations were distributed homogeneously across the entire protein. (4) Deleted amino acids (e.g. aromatic deletions) were replaced by weakly interacting amino acids (i.e. serines, threonines, glycines, and alanines). For each PLD, the identity of the weakly interacting residues used as replacements were chosen to preserve the percentages of the various types of weakly interacting amino acids present originally in the unmodified PLD. (5) To achieve a data set that spans a wide array of critical solution temperatures, as shown in *Figure 1D* and *Appendix 1—figure 6*.

## Validation of the Mpipi model for phase behavior of PLDs

The availability of quantitative experimental binodal for the hnRNPA1-IDR and various amino acid sequence variants *Bremer et al., 2022* has allowed us to test the accuracy of the Mpipi model to describe the phase diagrams of PLD solutions (*Joseph et al., 2021*). Here, we make such a test more stringent by extending the hnRNPA1-IDR validation set from 9 PLD variants to 21, selecting mutations spanning changes in aromatic, charged, polar, and neutral residues. The amino acid sequences of all the hnRNPA1-IDR variants considered are summarized in *Bremer et al., 2022* and are denoted using their established nomenclature: variants that mutate $n$ residues of type $X$ into $m$ residues of type $Z$ are labeled as $-nX + mZ$, and so forth. For more details on the nomenclature, see Methods.

For each hnRNPA1-IDR variant, we perform Direct Coexistence MD simulations, compute critical temperatures from the simulations, and compare them with the extracted experimental critical temperatures (*Figure 2A–D*). We use the fitting methodology outlined in Methods to estimate the critical solution temperatures from the experimental data. When comparing with experimental critical temperatures, our model achieves a Pearson correlation coefficient higher than 0.93 for all types

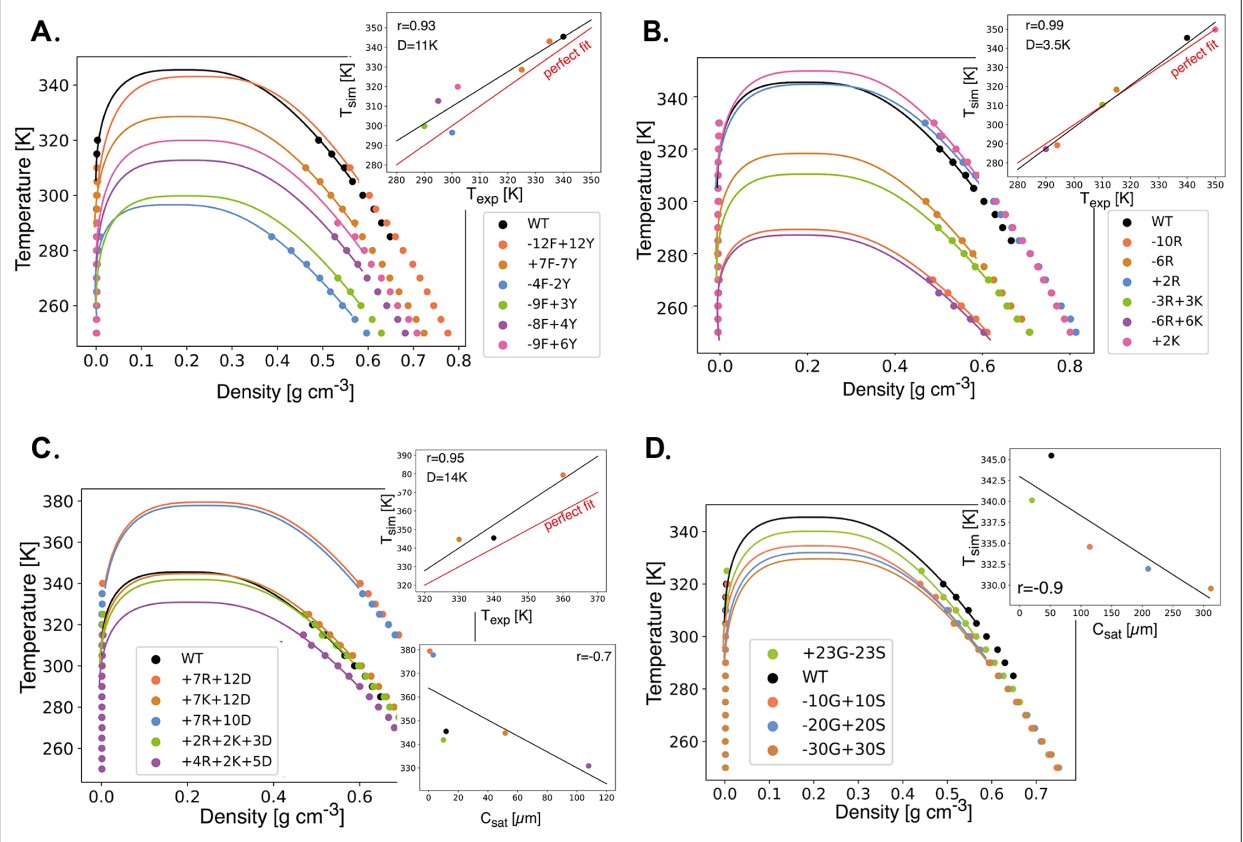

**Figure 2.** Mpipi model quantitatively captures phase behavior for a large set of heterogeneous nuclear ribonucleoprotein A1 (hnRNPA1) prion-like low complexity domain (PLD) mutants. Mutations are divided into (**A**) aromatic, (**B**) Arg/Lys, (**C**) charged, and (**D**) Gly/Ser sequence variants of the PLD of the protein hnRNPA1, and compared with experimental results. The hnRNPA1 PLD was used to validate the accuracy of the model, by extracting critical temperatures from experimental phase diagrams of a large set of these variants (**Bremer et al., 2022**). To assess how well the Mpipi model performed for PLDs, we compared critical temperatures obtained from simulations with the corresponding ones extracted from the experimental data for those variants that were available (depicted in the top right of each binodal, where *r* is the Pearson coefficient and *D* is the root mean square deviation between simulated and experimental values). For the variants where the experimental critical temperature could not be determined, we make a comparison between the critical temperature with the saturation concentration, since for these systems we expect them to be inversely correlated. The naming system for the variants is discussed in detail in the text.

of mutations analyzed. Additionally, for all cases, the root-mean-square deviation (*D*) between the experimental and simulated critical temperatures are $\leq 14$ Kelvin. Our validation set, which expands the range of protein variants originally tested (**Joseph et al., 2021**), highlights that the Mpipi potential can effectively capture the thermodynamic behavior of a wide range of hnRNPA1-PLD variants, and suggests that Mpipi is adequate for proteins with similar sequence compositions, as in the set of proteins analyzed in this study. In recent work by others (**Lotthammer et al., 2024**), Mpipi was tested against an experimental radius of gyration data for 137 disordered proteins and the model produced accurate predictions, which further suggests the applicability of Mpipi to a broad range of sequences.

## Large-scale data generation on critical parameters of PLD Phase separation

Next we focus on generating a dataset to extract the critical temperatures of the PLD variants. In the Direct Coexistence method, the diluted phase and the condensate are simulated in the same cuboid separated by an interface. The phase diagrams we compute explore the plane of temperature (vertical axis) versus density (horizontal axis). For each phase diagram, we run a set of about 20 Direct Coexistence simulations—each simulation explores the phase behavior of the system at a given temperature. For such temperature, if two phases are detected in our simulations, we measure a 'density profile:' the density of the proteins as a function of the long side of the simulation box (Bottom panel in **Figure 1C**). We then plot the two densities for each temperature we simulate and

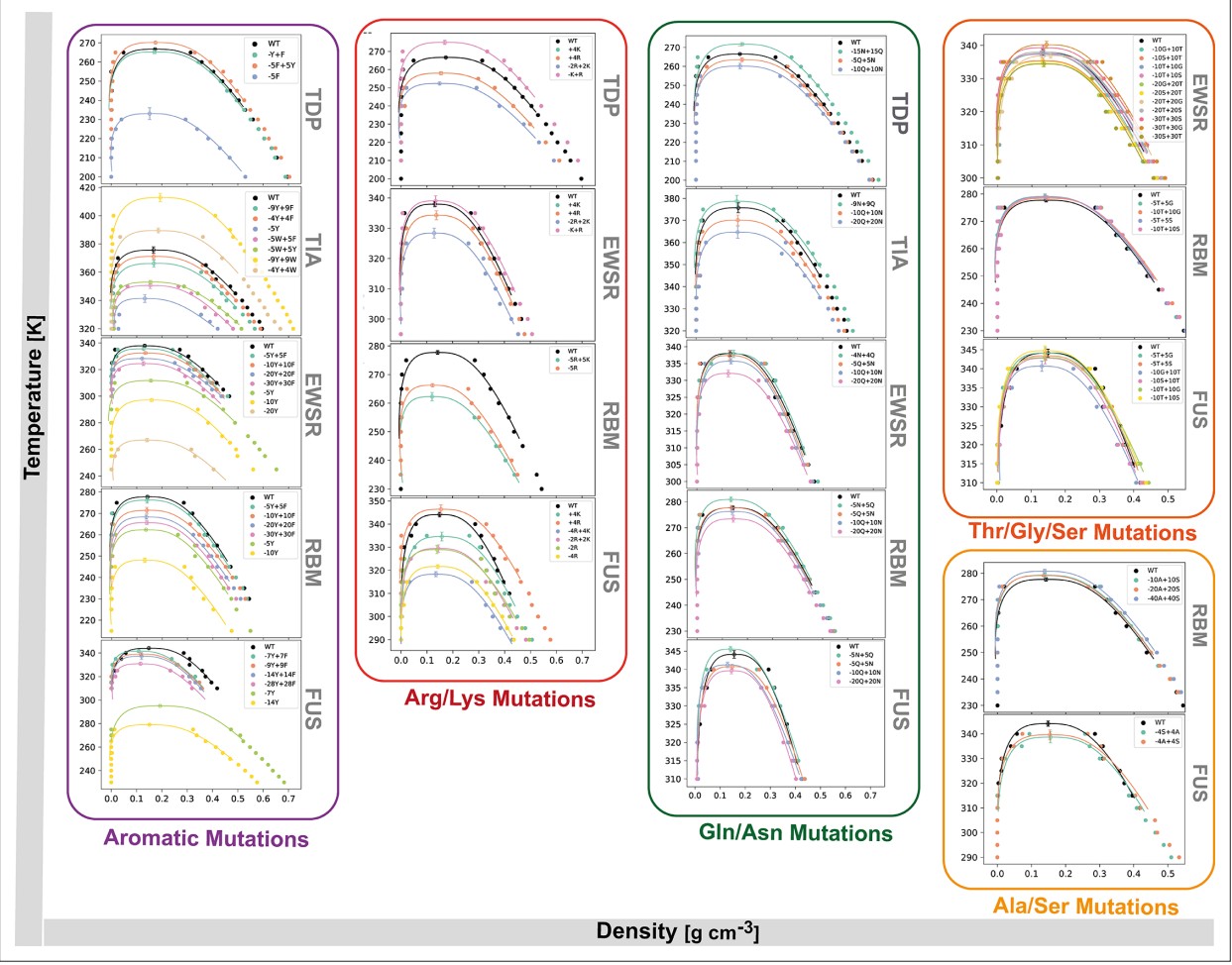

**Figure 3.** Complete set of 140 binodals, in the temperature versus density phase space, generated via large-scale molecular dynamics simulations. Binodals are grouped according to the mutation type (Aromatic, Arg/Lys, Gln/Asn (Polar), Ala/Ser and Thr/Gly/Ser (Neutral)). Each set is further grouped based on the prion-like low complexity domain (PLD) in question. The error shown in the fitted critical temperature corresponds to the uncertainty of the fit.

build the coexistence curve (also known as binodal), *Figure 3*. The coexistence curve shows the values of temperature and protein density for which phase separation will occur. The region above the coexistence curve is the 'one-phase region' where the temperature is too high for the enthalpic gain of the associative protein–protein interactions to overcome the entropic loss upon phase separation. The region below the coexistence curve, the 'two-phase region' represents temperatures for which the enthalpic gain due to protein–protein interactions is sufficiently high to favor phase separation into a condensed (protein-enriched) and a diluted (protein-depleted) liquid phase. The maximum in the coexistence curve is known as the critical point: the upper critical solution temperature (herein, critical solution temperature) and corresponding critical density. For temperatures higher than the critical solution temperature, phase separation is not observed.

The protein solutions that we investigate with our coarse-grained simulations must be sufficiently large, in terms of the number of copies of the same protein, to minimize finite size effects—for instance, to adequately sample bulk phenomena without dominating interfacial effects. At the same time, our simulations must be small enough to ensure computational convergence in a feasible timescale. Following testing of computational convergence for systems of different sizes (*Appendix 1—figure 8*) and MD trajectories of different timescales (*Appendix 1—figure 7*), we determined that a system composed of 64 protein replicas, with trajectories of 200 ns ensured convergence and reproducible results, while taking into consideration the trade-off between accuracy and computational demands. For the initial simulations of hnRNPA1 PLD to validate the model, we performed five

replicates to estimate the standard error of the density profiles. For subsequently designed systems, we employed an alternative method, running a long simulation divided into independent chunks. The first 200 ns of each trajectory were used for equilibration, as shown above, and excluded from the analysis. Following equilibration, snapshots were divided into chunks of 100 ns, ensuring no correlation between chunks. This method was validated with the hnRNPA1 PLD dataset, yielding consistent results with the random replicate-based approach. Standard error estimates were derived from the variability across these independent chunks or replicates. Our dataset of critical parameters is over an order of magnitude larger than previously reported studies (*Hennig et al., 2015*; *Bremer et al., 2022*; *Martin et al., 2020*; *Holehouse et al., 2021*; *Lin et al., 2015*; *Ryan et al., 2018*). This dataset (*Figure 3*) not only offers new insights into phase separation phenomena but also sets a new benchmark for data-driven research in this field.

## A data-driven scaling law approach for studying phase behavior of prion-like domains

Scaling laws can reveal key relationships between system variables, such as composition and critical parameters, providing a useful theory to predict and interpret the regulation of complex biological phenomena such as phase separation. In the context of PLD phase behavior, scaling laws would provide a quantitative theory to analyze how alterations in amino acid composition affect the critical solution parameters—a crucial aspect for understanding protein behavior in biological systems. Here, our objective is to discern which scaling laws can effectively capture the variations of PLD critical parameters as a function of sequence mutations. To accomplish this, we employ a systematic approach of varying the amino acid sequence of the PLDs considered, to determine how the phase behavior of PLDs changes with amino acid compositional perturbations.

For each type of mutation we analyze, we define and test three types of scaling laws. These scaling laws illustrate a different relationship between critical solution temperature changes and the number of amino acid mutations. The first and simplest scaling law is a linear relationship between these two quantities,

$$\Delta T_c = S_{X_1 \rightarrow X_2} N_{X_1 \rightarrow X_2} \tag{1}$$

where $\Delta T_c$ is the variation in the critical temperature between the mutated protein and the corresponding unmodified reference sequence, $X_1$ is an amino acid (e.g. Y, F, R, and so forth) and $N_{X_1 \rightarrow X_2}$ is the number of $X_1$ residues that have been mutated to $X_2$. Thus, $S_{X_1 \rightarrow X_2}$ in this case quantifies the variation in the critical solution temperature of the protein solution for a mutation of one $X_1$ residue to $X_2$. If the data is well represented by this first scaling law, it would imply that the effects of amino acid mutations on PLD phase behavior depend exclusively on the total number of mutations, but are not affected by other PLD parameters, such as the protein length. Thus, the first scaling law (*Equation 1*) is the simplest possible model we test here for how incremental mutations may collectively influence the phase behavior of PLDs.

Next, we define a second type of scaling law, which hypothesizes that the changes to the critical solution temperature upon mutation is proportional to the fraction of residues that are mutated, rather than their raw number. That is,

$$\Delta T_c = S_{X_1 \rightarrow X_2} \frac{N_{X_1 \rightarrow X_2}}{L} = S_{X_1 \rightarrow X_2} f_{X_1 \rightarrow X_2} \tag{2}$$

where, $L$ represents the number of total amino acids in the PLD, or protein length, and $f_{X_1 \rightarrow X_2} = \frac{N_{X_1 \rightarrow X_2}}{L}$ is the fraction of $X_1$ residues that have been mutated to $X_2$. Thus, this second scaling law hypothesizes that the effect of mutations on the critical solution temperature is more pronounced in shorter proteins (where $N_{X_1 \rightarrow X_2}$ mutations constitute a larger fraction of the protein) and less so in longer proteins (where $N_{X_1 \rightarrow X_2}$ mutations represent a smaller fraction of the total number of amino acids). The expectation is that because longer PLDs have higher valencies than shorter ones, the density of inter-molecular connections they form inside condensates is more robust towards perturbations than that of shorter PLDs (*Sanchez-Burgos et al., 2021*).

Finally, we define a third scaling law,

$$\Delta T_c = S_{X_1 \rightarrow X_2} \frac{N_{X_1 \rightarrow X_2}}{\sqrt{L}} \tag{3}$$

where the change in critical solution temperature is hypothesized to be directly proportional to the number of mutations ($N_{X_1 \rightarrow X_2}$) but inversely linear to $\sqrt{L}$. Like our second scaling law, this third relationship also presumes that longer PLDs form condensates with critical parameters that are less affected by mutations than shorter PLDs. However, by considering $\sqrt{L}$, instead of $L$ as in *Equation 2*, the expectation is that the buffering effects of protein length are more modest than those stated by *Equation 2*. Notably, the amino acid composition of PLDs have been explained by the stickers-and-spacers framework (*Choi and Pappu, 2020c*; *Ginell and Holehouse, 2022*). Aromatic and arginine residues have been proposed as stickers, contributing to the stability of the liquid network via associative interactions, yet some works argue that all residues can have strong effects in various contexts. Our data suggests that while a strict black-and-white delineation might not be appropriate, certain residues such as Tyr or Arg demonstrably have a larger impact on $T_c$ of PLDs. Conversely, other residues exhibit smaller effects, acting more like spacers interspersed among the stickers. *Equation 3* suggests that as a protein lengthens, the number of residues that can establish sufficiently strong inter-protein interactions grows too; yet, it grows much more conservatively than $L$ because a fraction of $L$ is composed of spacers.

With these scaling laws as ansätz, we set out to uncover trends in the dataset we built from our MD simulations of 140 variants of different PLDs. Adopting these laws as foundational hypotheses, we explore the relationship between alterations in protein sequence and changes in critical solution temperatures, offering insights into the molecular drivers of PLD phase separation.

## Aromatic mutations exhibit a linear correlation between mutation fraction and critical solution temperatures

Aromatic amino acids, namely tyrosine (Tyr, Y), phenylalanine (Phe, F), and tryptophan (Trp, W), are widely regarded as essential contributors to phase separation of PLDs, due to their ability to form π–π and cation–π contacts (*Vernon et al., 2018*; *Das et al., 2020*; *Bremer et al., 2022*; *Martin et al., 2020*; *Krainer et al., 2021*; *Fisher and Elbaum-Garfinkle, 2020*; *Fossat et al., 2021*; *Dyson et al., 2006*; *Andrew et al., 2002*). Among aromatics, experiments and simulations agree that Tyr is a stronger contributor to the stability of condensates than Phe (*Bremer et al., 2022*; *Krainer et al., 2021*; *Wang et al., 2018*; *Qamar et al., 2018*), and that both the composition and the patterning of aromatic residues in PLDs have an impact on the condensate critical parameters and its material state (*Holehouse et al., 2021*; *Li and Jiang, 2022*; *Alshareedah et al., 2023*; *Tejedor et al., 2023*).

Aromatic residues are highly abundant in PLDs, with the fraction of aromatic residues in the PLD variants analyzed in this study ranging from 0.19 to 0.43 (*Figure 1B*). Furthermore, because different aromatics are unequal contributors to PLD phase behavior, we analyzed the impact of Tyr, Phe, and Trp mutations separately.

To quantify the impact of mutating Tyr to Phe in shifting the critical solution temperatures of our set of PLDs—i.e., determining which of our proposed scaling laws best describes the data—we designed 17 protein mutants that varied the Tyr to Phe ratio in the family of PLDs and computed their phase diagrams (*Figure 4A*). Our simulations reveal that Tyr is stronger than Phe as a driving force for phase separation for the entire PLD family considered, a finding that replicates the results of *Bremer et al., 2022* for hnRNPA1 PLD and confirms that it extends to other PLDs. By analyzing the change of critical solution temperature with respect to the unmodified PLD (or wild-type, WT) against the number of mutations, our results reveal a linear relationship between the number of Tyr to Phe mutations and the change in the critical temperature of the system; our first scaling law, $S_{Y \rightarrow F}$, *Equation 4*. Here, $\Delta T_c$ represents the temperature difference between the mutated protein and its wild-type sequence, and $N_{Y \rightarrow F}$ represents the number of Tyr residues that were mutated to Phe in the sequence.

$$S_{Y \rightarrow F} = \frac{\Delta T_c}{N_{Y \rightarrow F}} = -0.40 \pm 0.04 \tag{4}$$

Note that, by definition $S_{Y \rightarrow F} = -S_{F \rightarrow Y}$, since $N_{Y \rightarrow F}$ is defined as positive for Tyr to Phe mutations and negative for Phe to Tyr mutations, or, in other words, $N_{Y \rightarrow F} = -N_{F \rightarrow Y}$. For our data set, we find $S_{Y \rightarrow F}$ to be $-0.40 \pm 0.04$, where the error given is the standard error. The way to interpret this result is that a single mutation of a Tyr residue to a Phe residue in a given PLD protein would result in a 0.4 K variation in the critical solution temperature for phase separation. Such a modest variation due to a single-point mutation is smaller than the typical error associated with estimating the absolute value

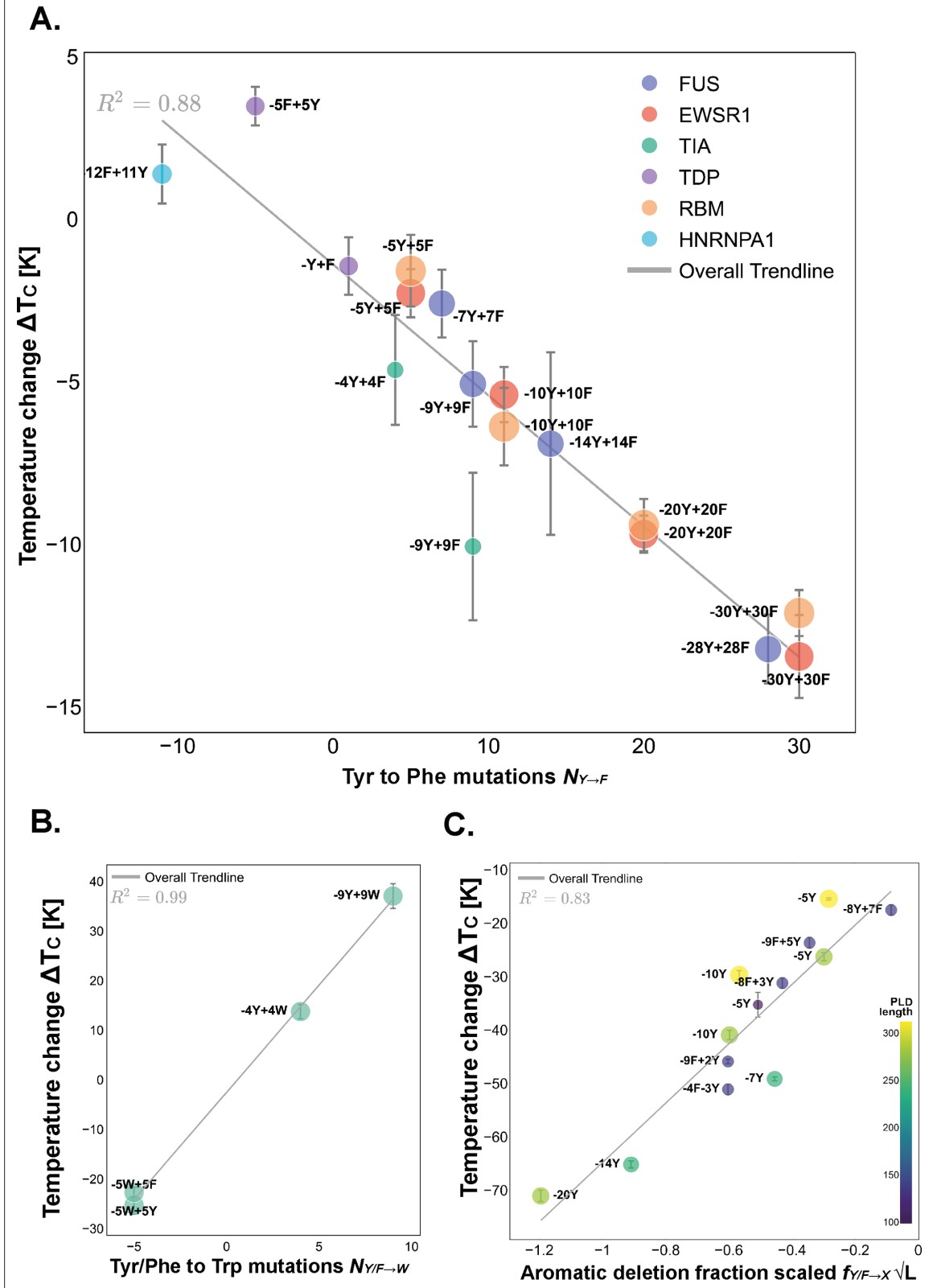

**Figure 4.** Mutations in aromatic amino acids have strong effects on the critical solution temperature of prion-like low complexity domains (PLDs). (**A**) Tyr to Phe mutations. The number of Tyr mutated to Phe (x-axis) versus the change in critical temperature, (y-axis), computed as the critical temperature of the variant minus that of the corresponding PLD wild-type sequence. The trendline defines the simplest, dominating scaling law for this mutation. The $R^2$ value of 0.88 shows the agreement of our data to the scaling law defined by *Equation 4*. Errors in critical temperature are computed via error

*Figure 4 continued on next page*

*Figure 4 continued*

propagation, considering the uncertainty of the binodal fit. (**B**) Tyr or Phe to Trp mutations. In this case, the overall trendline, defined by *Equation 5* has a $R^2$ of 0.99. (**C**) Analysis of the variants involving mutations of aromatic residues to uncharged, non-aromatic amino acids, a.k.a. aromatic deletions. In the y-axis, the change in the critical temperature of the variant to that of the wild-type sequence, and in the x-axis, a renormalized measure of mutations: the fraction of aromatic residues mutated times the $\sqrt{L}$, to account for the competing physical interactions between PLDs. Error bars indicate the standard error associated with replicas of the simulation.

of the critical solution temperature itself using residue-resolution coarse-grained Direct Coexistence simulations (approximately 3 K). Unambiguously measuring the value of the critical solution temperature with sub-Kelvin accuracy also poses a considerable challenge in experiments (*Alberti et al., 2019*; *Ray and Buell, 2024*). However, we can uncover statistically significant trends by analyzing the behavior of the critical solution temperature in a large set of PLDs with diverse numbers of mutations rather than a single case. In particular, our analysis reveals that the impact of Tyr to Phe mutations is modest and the effect of multiple mutations can be described adequately as cumulative.

In addition to these analyses, the role of Tyr/Phe to Trp mutations, albeit Trp being a rare residue in prion domains, was investigated in TIA1 due to the presence of Trp in the wild-type PLD, *Figure 4B*. The simplest scaling law that describes this mutation (*Equation 5*) is fundamentally equal to the law describing Tyr to Phe mutations. However, a notable distinction is that the $S_{F/Y \to W}$ scaling constant for our data is approximately $4.3 \pm 0.1$, which is one order of magnitude greater than $S_{Y \to F}$, consistent with the much stronger $\pi$–$\pi$ contacts established by Trp. It is important to note that, in this case, similarly to Try to Phe mutations, multiple ansätze align closely with the observed data, which underscores the fact that, in the context of these mutations, the size of the side chains do not significantly alter phase behavior, as the predominance of aromatic perturbations exerts a dominating effect.

$$S_{F/Y \to W} = \frac{\Delta T_c}{N_{F/Y \to W}} = 4.3 \pm 0.1 \tag{5}$$

Across the whole family of PLDs analyzed, our data reveals that changes in aromatic amino acid identity impacts the critical solution temperatures modestly (<1 K per mutations involving F and Y) to strongly (4 K for mutation involving Trp) and independently of the protein length. Such a result is consistent with the mutations in aromatic identity preserving the total number of associative interactions that the PLDs can establish, and hence, the density of molecular connections in the condensed liquid network. That is, aromatic mutations are expected to influence the energetics and lifetimes of the multivalent interactions among PLDs, but not the number of intermolecular contacts they form.

## Strong critical temperature reduction due to aromatic deletions is buffered by protein length

We next examine the role of replacing aromatic residues (Tyr and Phe) with uncharged, non-aromatic residues—such as Gly, Ser, Gln, Ala, and Asn—which are prevalent in PLDs. The mutations were designed to preserve amino acid sequence compositional biases by replacing Tyr and Phe with uncharged, non-aromatic residues in the same proportions as they appear in the wild type of each sequence, i.e., maintaining the relative amounts of each type of residue found in the sequence. All amino acid sequences are available in Appendix 1.

Our simulations show that aromatic deletions strongly reduce the critical solution temperature of the PLD solutions, *Figure 4C*. The best scaling law from our ansätz to describe the data is *Equation 3*, which involves the square root of the PLD length. That is, $S_{Y/F \to X}$, where $X$ represents any uncharged, non-aromatic residue.

$$S_{Y/F \to X} = \frac{\Delta T_c \sqrt{L}}{N_{Y/F \to X}} = -56 \pm 5 \tag{6}$$

Here, once again, $N_{Y/F \to X}$ is the number of mutations, and $L$ is the length of the PLD. For our data set, the value of the scaling constant $S_{Y/F \to X}$ is $-56 \pm 5$. To interpret this result, let us consider a model PLD of two hundred amino acids ($L = 200$). If we mutate one Tyr to an uncharged, non-aromatic amino acid, the change in the critical temperature—computed via the formula $S_{Y/F \to X} \times \frac{N_{Y \to X}}{\sqrt{L}}$ is approximately -4 K. Such a value is consistent with the dominance of $\pi$–$\pi$ and cation–$\pi$ contacts in PLD

phase separation reported experimentally (*Vernon et al., 2018*; *Das et al., 2020*; *Bremer et al., 2022*; *Martin et al., 2020*; *Krainer et al., 2021*; *Fisher and Elbaum-Garfinkle, 2020*; *Fossat et al., 2021*; *Dyson et al., 2006*; *Andrew et al., 2002*). As stated above, aromatic residues behave as associate hubs within PLDs condensates. When these aromatic residues are replaced with uncharged non-aromatic amino acids the network of molecular connections in the liquid are strongly perturbed, as predicted earlier (*Espinosa et al., 2020*). To observe this in more detail, we computed select contact maps (see *Appendix 1—figure 10*)—notably, mutations that removed aromatic amino acids strongly altered the primary sites of interactions.

Our results reveal that the length of the PLDs strongly influences how the critical parameters of PLD solutions change in response to aromatic deletions. The effect of mutations is stronger in shorter proteins because each aromatic deletion represents a higher percentage of the total number of associative residues in the PLD, than in longer proteins. Longer proteins have higher valencies and redundant 'stickers' that facilitate a network rearrangement.

Our results highlight the delicate balance between amino acid composition and protein length in determining the phase behavior of PLDs, providing a deeper understanding of the molecular basis of protein phase transitions.

## Arginine mutations drastically alter condensate stability

The distribution and balance of charged amino acids—i.e., aspartic acid (Asp, D), glutamic acid (Glu, E), histidine (His, H), arginine (Arg, R), and lysine (Lys, K)—within an IDR affects its solubility, conformational ensemble, and the range of stability of the ensuing condensates it might form. PLDs are not particularly rich in charged residues: in the wild-type PLDs of our dataset, the fraction of charged residue ranges from 0.02 to 0.1 (*Figure 1B*), and the fraction of positively charged amino acids is narrowly distributed around 0.015, except for hnRNPA1, which has a value of 0.09 (*Figure 1B*). However, despite their low prevalence in PLDs, charged residues, in particular Arg, have been shown to be important contributors to the stability of a wide-range of biomolecular condensates (*Bremer et al., 2022*; *Fisher and Elbaum-Garfinkle, 2020*; *Hong et al., 2022*; *Greig et al., 2020*; *Strom et al., 2017*).

While both Arg and Lys are positively charged amino acids at physiological pH, it has been demonstrated that they have unequal roles in the stability of biomolecular condensates (*Bremer et al., 2022*;

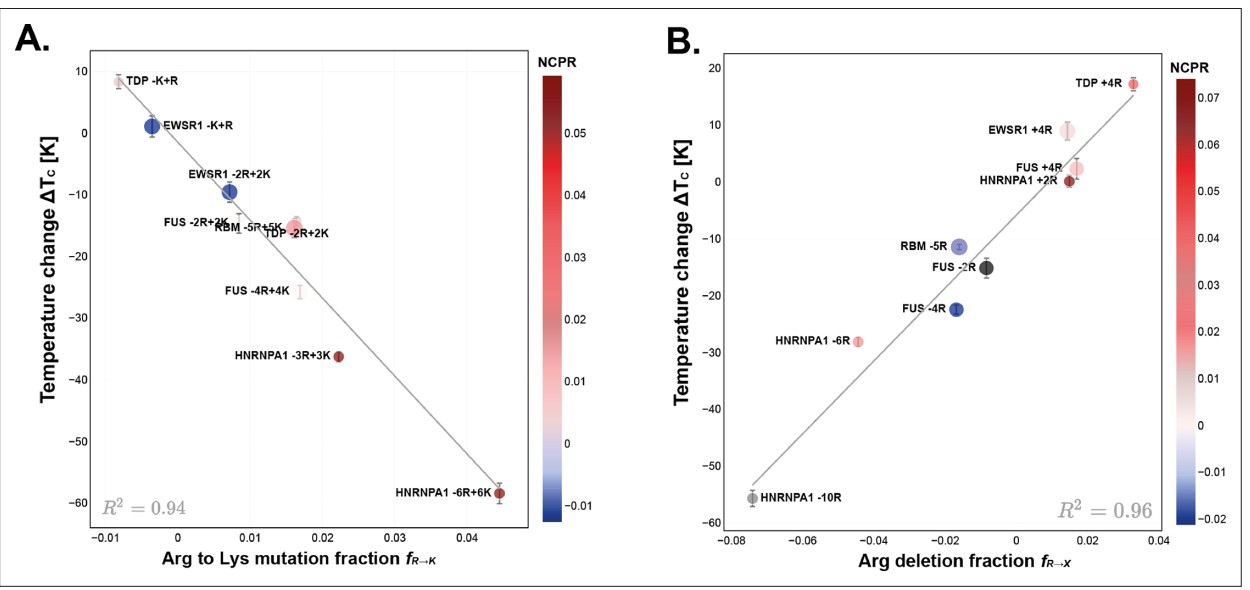

**Figure 5.** Effects of Arg mutations on the phase transition temperature of prion-like low complexity domains (PLDs). (**A**) Analysis of the critical solution temperature of PLDs with Arg to Lys mutations. The number of Arg to Lys mutations divided by PLD length (x-axis) versus the change in the critical solution temperature, that is, $T_c(\text{variant}) - T_c(\text{WT})$ (y-axis). The trendline, representing *Equation 7*, fits the data with a $R^2 = 0.94$, and defines the stability measure of this perturbation. (**B**) Analysis of the critical solution temperature of PLDs with Arg mutations to uncharged, non-aromatic amino acids that maintain the wild-type (WT) compositional percentages (mainly Gly, Ser, and Ala). For more details see Appendix 1. Arg deletion fraction (x-axis) versus $T_c(\text{variant}) - T_c(\text{WT})$ (y-axis). The trendline, which fits the data with $R^2 = 0.96$, defines the stability measure, represented in *Equation 8*.

*Greig et al., 2020*; *Fisher and Elbaum-Garfinkle, 2020*; *Mitchell et al., 2000*; *Qamar et al., 2018*).
Both residues can form charge–charge and cation–π associative interactions, but the case of Arg
is unique: the $sp^2$-hybridized planar guanidinium group of Arg confers it a special capacity to form
strong hybrid π–π/cation–π contacts with the π-electrons of aromatic rings (*Krainer et al., 2021*).
Thus, to quantify the impact of Arg in the range of stability of PLD condensates, we designed mutants
that replaced Arg with Lys and vice-versa (*Figure 5*).

The best fitting scaling law associated with mutating Arg to Lys is inversely dependent on the
fraction of mutations, rather than the number of residues mutated (*Equation 7*). The resulting scaling
constant for this particular mutation, i.e., $S_{R \to K}$, has a value of $-1300 \pm 100$, indicating that for a typical
PLD of $L = 200$, the mutation of a single Arg into Lys results in a reduction of the critical temperature
of the system by 6.5 K. Hence, replacing Arg with Lys is a destabilizing mutation of PLD condensates
in agreement with the wide-body of experimental evidence (*Bremer et al., 2022*; *Greig et al., 2020*;
*Fisher and Elbaum-Garfinkle, 2020*; *Mitchell et al., 2000*; *Qamar et al., 2018*).

$$S_{R \to K} = \frac{\Delta T_c}{f_{R \to K}} = -1300 \pm 100 \qquad (7)$$

The length dependency of this particular scaling law suggests that the impact of Arg to Lys
mutations on the phase behavior of PLDs primarily depends on the relative concentration of these
mutations (mutation fraction). Additionally, it proposes that longer sequences may exhibit increased
resilience against Arg to Lys mutations. This resilience is likely attributed to the greater separation of
charge within the longer sequences.

To further analyze the effect of the charge of the PLDs in this set of mutants, in *Figure 5* we include
a scalebar showing the net charge per residue (NCPR). Mutants that present higher NCPR present
the biggest decrease in their phase-separating propensities (*Figure 5A*), which is consistent with the
expectation that condensates with a neutral overall charge are more energetically stable (*Bremer
et al., 2022*; *Das and Pappu, 2013*; *Cohan et al., 2022*).

When we replace Arg with uncharged, non-aromatic residues, the resulting scaling law also
depends inversely on the fraction of mutations, with a value of $-640 \pm 50$, amounting to a reduction
of 3.2 K in $T_c$ per Arg deletion for a model PLD of length $L = 200$.

$$S_{R \to X} = \frac{\Delta T_c}{f_{R \to X}} = -640 \pm 50 \qquad (8)$$

Therefore, our results reveal that it is almost twice more destabilizing to mutate Arg to Lys than to
replace Arg with any uncharged, non-aromatic amino acid, namely, Gly, Ser, or Ala. Mutating Arg to
an uncharged non-aromatic amino acid is less destabilizing than mutating it to Lys likely because the
charge-reduced variants decrease the cation–cation repulsion of the PLDs, with respect to the Arg-
rich or Lys-rich systems. Furthermore, mutating Arg to Lys decreases the enthalpic gain for condensate
formation not only via the replacement of stronger Arg-based associative interactions for weaker Lys-
based ones but also by increasing the overall solubility of the Lys-rich PLD (*Fossat et al., 2021*; *Zeng
et al., 2022*), given that the hydration free energy of Arg is less favorable than that of Lys (*Fossat
et al., 2021*; *Krainer et al., 2021*; *Hong et al., 2022*).

## Polar amino acids subtly modulate PLD phase behavior

Uncharged, non-aromatic amino acids, including Gly, Ala, proline (P, Pro), Ser, and Thr, are commonly
found in PLDs. Because they generally form weaker residue-residue interactions and present smaller
excluded volumes than aromatics, they are often classified as spacers that contribute to maintaining
the solubility of proteins and material states of condensates (*Holehouse et al., 2021*; *Choi and
Pappu, 2020c*). These amino acids can also have other functions in proteins, and the role they play as
depends on their chemical make-up, but also on the specific context in which they are found (*Bremer
et al., 2022*; *Martin and Mittag, 2018*; *De Sancho, 2022*).

Here, we investigate the role of glutamine (Gln, Q) and asparagine (Asp, N). Both Gln and Asn
are polar amino acids, hence their charge distribution is more asymmetric, presenting a significant
dipole moment. Consequently, they can establish transient ion–dipole, dipole–dipole and π–dipole
interactions with other residues. While the Mpipi model does not account explicitly for polarity, it
is worth noting that it was parametrized based on bioinformatics data, which implicitly incorporates

this aspect. Thus, our study allows for the observation of emergent behaviors that stem from these inherent characteristics.

We designed mutations that vary the relative amounts of Gln and Asn, and examined trends in their phase-separating propensity. Increasing the number of Gln per unit length by mutating Asn to Gln increases the ability of the system to phase separate, likely due to Gln having a higher dipole moment than Asn, which is approximated in our model via marginally stronger non-ionic interactions formed by Gln over Asn (**Figure 6A**).

$$S_{N \to Q} = \frac{\Delta T_c}{f_{N \to Q}} = 50 \pm 5 \tag{9}$$

The value of the scaling constant $S_{N \to Q}$ is $50 \pm 5$, which is significantly lower than that of mutations of residues with a net charge or aromatic deletions (rounding 1300, for example, for arginine to lysine mutations). This result indicates that mutations of Asn to Gln or vice versa have a smaller impact in the critical parameters than aromatic deletions or mutations of charged residues.

Finally, we also explored the role of Ser, Gly, Thr, and Ala on the critical parameters of the condensate. While Ser and Thr are also polar amino acids, their dipole moment is significantly smaller than that of Gln and Asp. Consistently, the scaling constants involving mutations Ser and Thr mutations—i.e., $S_{S \to T}$— are considerably smaller (**Figure 6B and C**) overall. A comparable pattern is noted for $S_{G \to T}$, with G expected to exhibit significantly lower polarity compared to Gln and Asp.

$$S_{S \to T} = \frac{\Delta T_c}{N_{S \to T}} = -0.09 \pm 0.03 \tag{10}$$

$$S_{G \to T} = \frac{\Delta T_c}{N_{G \to T}} = -0.08 \pm 0.02 \tag{11}$$

Regarding Ala, we also tested its relative role with respect to Ser.

$$S_{A \to S} = \frac{\Delta T_c}{N_{A \to S}} = 0.16 \pm 0.05 \tag{12}$$

Amino acids with larger side chains, such as Thr, have a larger excluded volume than those with smaller side chains, such as Gly or Ser. The corresponding value of the scaling constants $S_{S \to T}$ and $S_{G \to T}$ suggests that in the case of Thr mutations, mutations of neutral, non-aromatic amino acids to those with larger excluded volumes inhibit phase separation of the PLD by lowering the corresponding critical temperature, while mutations to those amino acids with smaller excluded volumes have an opposite effect. This is likely due to the increased flexibility of the protein backbone, allowing the chain to adopt multiple conformations that promote multivalent contacts. Importantly, these trends cannot be anticipated by inspecting the parameters of the Mpipi model; instead, they emerge from examining the behavior of the collective system.

On the other hand, for mutations involving Ser, Gly, and Ala—all small amino acids with minor side chain size variations—the impact on phase separation likely stems predominantly from changes in solvation. Solvation plays a key role in phase separation since it can help balance the driving forces of this process (**Ahlers et al., 2021**; **Nomoto et al., 2021**). While hydrophobic regions of the protein, with low solvation free energies, tend to form aggregates to reduce interactions with the solvent environment; hydrophilic regions promote interactions with water. Whether these regions inhibit or promote phase separation is likely due to a balance of opposing effects. For instance, hydrophilic regions can form energetically favorable contacts with solvent, promoting dispersion and suppressing phase separation. However, when PLDs are recruited to condensate the release of water molecules can help to off-set the entropic cost of sequestering hydrophobic regions within a protein droplet, thereby facilitating phase separation.

Specifically, mutations from Gly to Ser (not shown) increase the size of the side chain while also enhancing solvation, while mutations from Ala to Ser do not significantly change the size of the side chain but significantly increase the protein solvation. In the first case, the critical parameters of phase separation remains virtually unchanged, only presenting a very minor decrease. In the case of Ala to Ser mutations, phase separation is slightly enhanced. Both scaling laws $S_{G \to S}$ and $S_{A \to S}$ reflect such an effect, and underlie the context-dependent nature of these mutations.

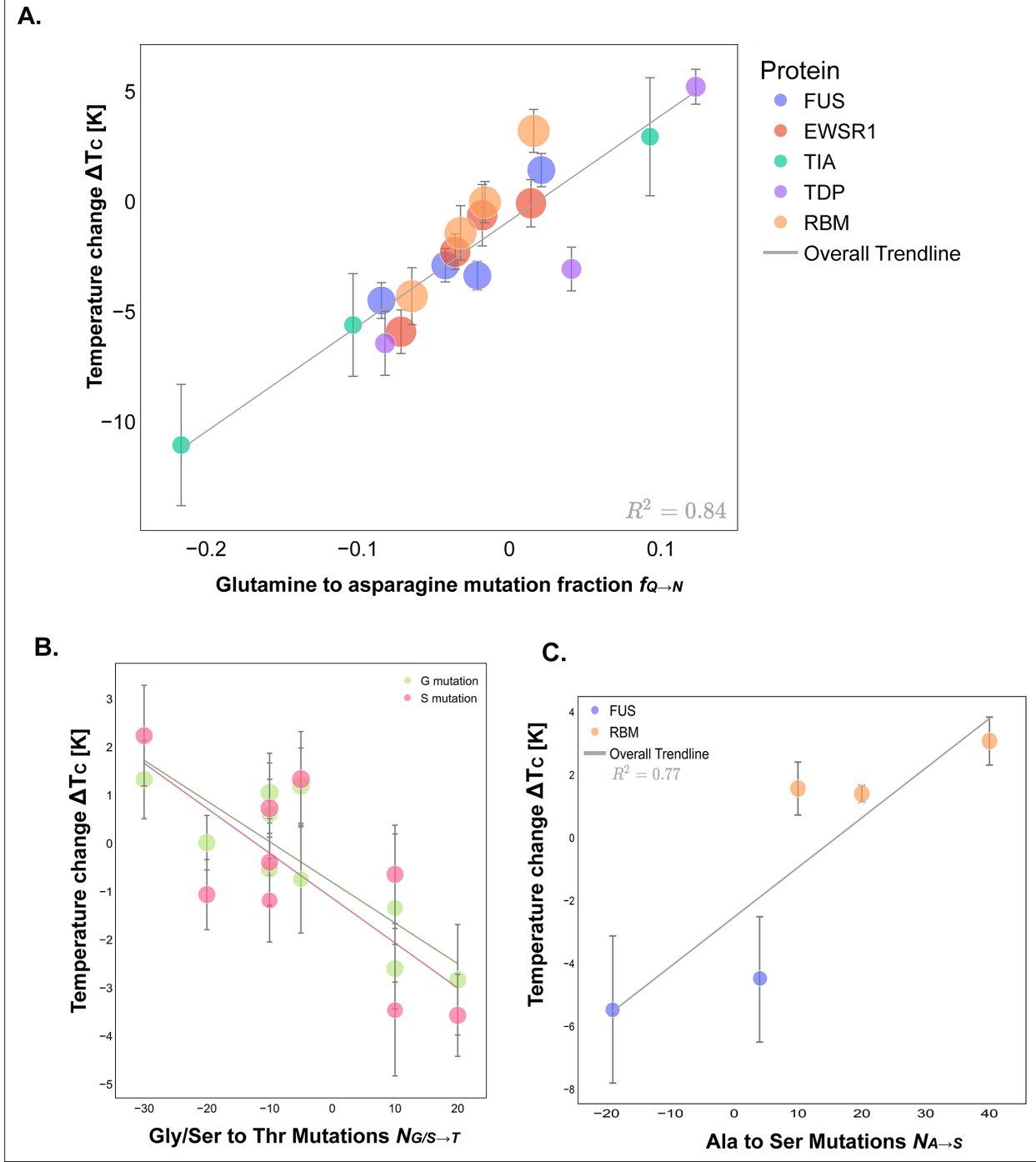

**Figure 6.** Polar and neutral amino acid mutations show subtle modulation of the critical solution temperatures of prion-like low complexity domains (PLDs). (**A**) Analysis of the critical solution temperature of PLDs with Asn to Gln mutations. The number of Gln to Asn mutations renormalized by dividing by PLD length (x-axis) versus the temperature change $T_c$(variant) $- T_c$(WT) (y-axis). The trendline, which fits the data with a $R^2 = 0.84$, fits the scaling law in *Equation 9*. (**B**) Analysis of the critical solution temperature of PLDs with Gly/Ser to Thr mutations (x-axis) versus the temperature change $T_c$(variant) $- T_c$(WT) (y-axis). The trendlines, representing *Equations 10 and 11*, fit the data with a $R^2 = 0.7$. (**C**) The number of Ala to Ser mutations (x-axis) versus the temperature change $T_c$(variant) $- T_c$(WT). The trendline, *Equation 12*, fits the data with a $R^2 = 0.77$.

Overall, mutations of unchanged, non-aromatic amino acids seem to have an impact on phase separation due to two competing effects: a change in side chain size and a change in solvation. However, it is important to note that the magnitude of the scaling laws that govern these mutations is minimal, and thus, likely extremely sensitive to small fluctuations such as changes in the system and

solution conditions. Notably, these findings are consistent with experiments that report small changes in phase separation propensity for mutations involving neutral, non-aromatic residues (*Bremer et al., 2022*; *Wang et al., 2018*).

## Beyond composition: Patterning considerations

Beyond composition, the relative positioning of amino acids in the sequence, i.e., sequence pattern, also plays a role in protein folding, function, and, in this case, phase behavior (*Martin et al., 2020*; *Zheng et al., 2020*; *Weiner et al., 2021*). Due to the sequence heterogeneity of PLDs, and, more generally, IDPs, it is particularly complex to define adequate patterning measures (*Cohan et al., 2022*; *Huihui and Ghosh, 2021*; *Ghosh et al., 2022*; *Rana et al., 2021*). In this work, we have taken advantage of our large data set to analyze the impact of amino acid sequence patterning by adopting a patterning order parameter, $\sigma_{\text{aro}}$, that has been designed to describe the patterning of aromatic residues in such IDPs (*Martin et al., 2020*; *Martin et al., 2016*; *Das and Pappu, 2013*).

To compute $\sigma_{\text{aro}}$, we divide the amino acid sequence into peptide segments of equal length ($l$), compute a measure of sequence aromatic asymmetry for each segment, and finally sum the values of such measure over all the segments. Consistent with literature (*Martin et al., 2020*; *Martin et al., 2016*; *Das and Pappu, 2013*), we take $l$ to be 5 and 6. $\sigma_{\text{aro}}$ is then given by:

$$\sigma_{\text{aro}} = \left( \sum_{\text{segments}} \frac{\sigma_i}{\sigma_{\text{max},i}} \right) \frac{N_{\text{stickers}}}{L} \tag{13}$$

where $L$ is the total number of amino acids in the protein domain, $N_{\text{stickers}}$ the number of sticker residues—i.e., charged or aromatic residues—and $\sigma_i$ has the functional form

$$\sigma_i = \left( \left( 2\frac{N_{\text{aro},i}}{l_i} - 1 \right)^2 - \sigma_T \right)^2. \tag{14}$$

Here, $N_{\text{aro},i}$ is the number of aromatic residues in the -th segment, $l_i$ the segment length, and $\sigma_T$ is a function of the total number of aromatic residues in the sequence $N_{\text{aro}}$:

$$\sigma_T = \left( 2\frac{N_{\text{aro}}}{L} - 1 \right)^2 \tag{15}$$

In turn, $\sigma_{\text{max},i}$ is the measure of aromatic amino acid distribution for an artificial sequence with all the aromatic amino acids concentrated in part of the chain.

Our results, presented in *Figure 7*, indicate that the sequences with the most dispersed aromatic residues (i.e. low values of $\sigma_{\text{aro}}$) phase separate more readily. Dispersed residues result in greater effective valency of a PLD, leading to a higher probability of forming a percolated network structure to sustain the condensate. Additionally, this result demonstrates that the patterning order parameter, $\sigma_{\text{aro}}$, can be used to predict the phase-separating propensity of a PLDs as the function of aromatic residue distribution. This result is in agreement with previous studies that demonstrate the importance of aromatic residue patterning on the phase behaviour of PLDs (*Martin et al., 2020*; *Holehouse et al., 2021 Holehouse et al., 2021*).

## Discussion

Computer simulations have advanced our ability to map emergent properties of protein solutions to the chemical make-up of their individual biomolecules. For instance, modern residue-resolution protein coarse-grained models (*Regy et al., 2021*; *Joseph et al., 2021*; *Dignon et al., 2018*; *Tesei et al., 2021*; *Mammen Regy et al., 2021*; *Das et al., 2020*; *Latham and Zhang, 2020*) can now be used to quantify the modulation of protein phase behavior as a result of amino acid sequence mutations of their native proteins with near-quantitative agreement with experiments (*Joseph et al., 2021*; *Mammen Regy et al., 2021*; *Tesei et al., 2021*). In this study, we exploit the accuracy and efficiency of our protein model, Mpipi (*Joseph et al., 2021*), to compute and analyze the temperature-vs-density phase diagrams of a large set of PLDs—140 variants of the PLDs of proteins hnRNPA1, TDP43, FUS, EWSR1, RBM14, and TIA1. Collectively, our set of phase diagrams reveals conserved scaling laws that

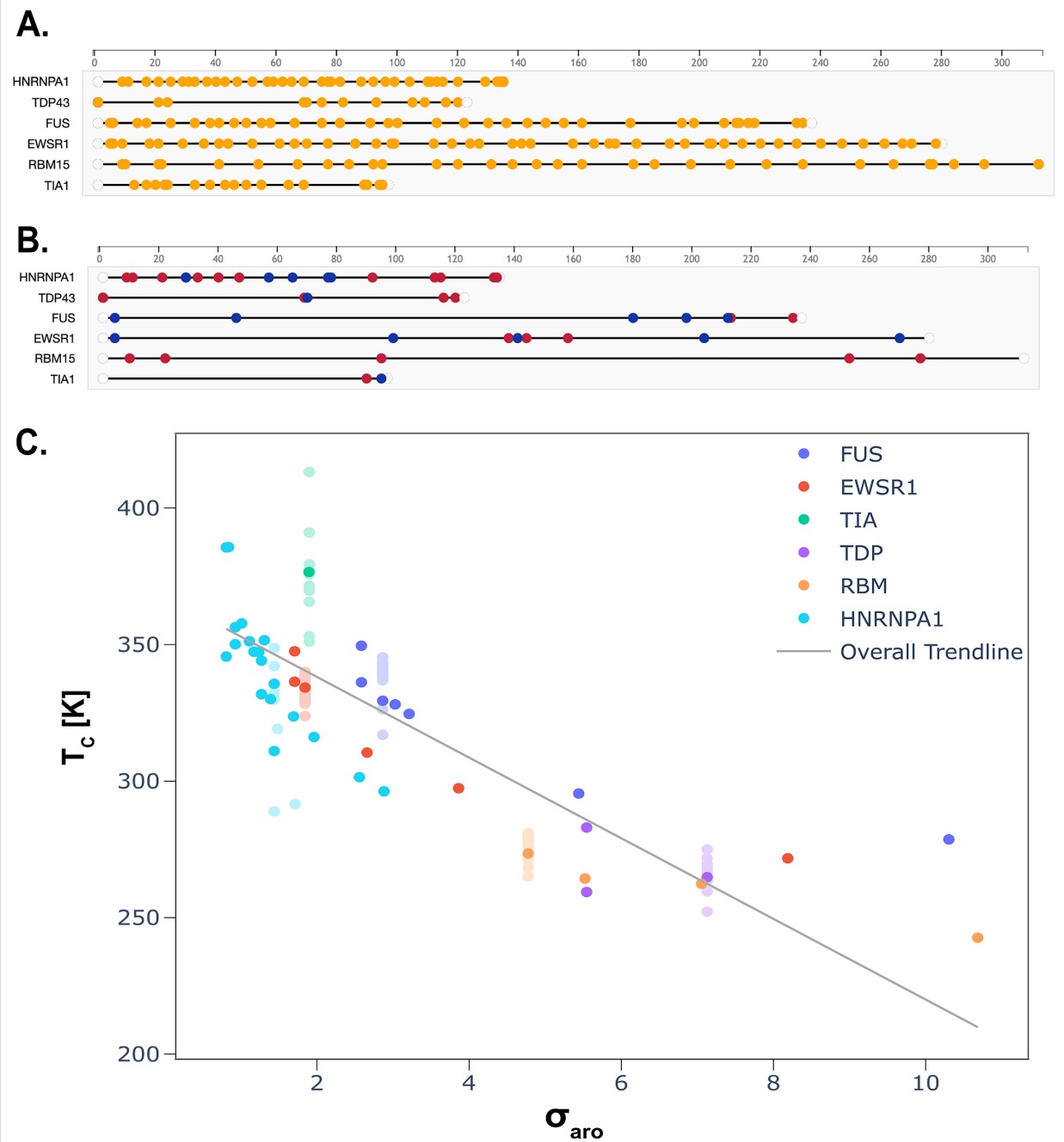

**Figure 7.** Prion-like domains featuring sequences with dispersed aromatic residues show greater propensities for phase separation. (**A**) Representation of the aromatic amino acids present in the wild-type sequences of each prion-like low complexity domain (PLD) considered. (**B**) Location of the positively charged (red) and negatively charged (blue) amino acids present in the wild-type sequences of each PLD considered. (**C**) Analysis of the critical solution temperature versus the $\sigma_{aro}$ order parameter, defined in *Equation 13*. Those groups of variants with the same $\sigma_{aro}$ value show their average critical temperature highlighted, with lighter symbols for other data points. For lower values of $\sigma_{aro}$, the critical solution temperature is higher, and vice versa, indicating that those variants with a more homogeneous distribution of aromatic amino acids have condensates that are stable at higher temperatures.

quantitatively predict the change in the critical solution temperatures of single-component PLD solutions as a function of the number and type of amino acid mutations of the PLDs.

To facilitate the interpretation of the scaling laws derived in our study, we have organized the data in *Table 1*, which illustrates the effect of single point mutations on the critical solution temperatures of the family of PLDs, as a function of the PLD length.

**Table 1.** Predicted effect of single point mutations on the critical temperature, in Kelvin, of prion-like low complexity domains (PLDs), as a function of their length, $L$.

| L | Y→F | F/Y→W | F/Y→X | R→K | R→X |
|---|---|---|---|---|---|
| 100 | −0.4±0.0 | 4.3±0.1 | −5.6±0.5 | −13±1 | −6.4±0.5 |
| 200 | −0.4±0.04 | 4.3±0.1 | −3.9±0.4 | −6.5±0.5 | −3.2±0.3 |
| 300 | −0.4±0.04 | 4.3±0.1 | −3.2±0.3 | − 4.3±0.3 | −2.1±0.2 |

| L | N→Q | S→T | G→T | A→S |
|---|---|---|---|---|
| 100 | 0.5±0.05 | −0.09±0.03 | −0.08±0.02 | 0.16±0.05 |
| 200 | 0.25±0.03 | −0.09±0.03 | −0.08±0.02 | 0.16±0.05 |
| 300 | 0.17±0.02 | −0.09±0.03 | −0.08±0.02 | 0.16±0.05 |

Consistent with experimental evidence and computational predictions, interactions involving aromatic residues—namely, $\pi$–$\pi$ and cation $-\pi$— are dominant in stabilizing condensates formed by PLDs (**Wang et al., 2018**; **Bremer et al., 2022**; **Joseph et al., 2021**). Our simulations reveal the largest decrease in the critical solution temperature of a PLD condensate occurs when an aromatic residue is replaced by an uncharged, non-aromatic residue. The scaling law for such aromatic deletions ($56\,\mathrm{K}/\sqrt{L}$) predicts that the destabilizing effect of replacing an aromatic with a non-aromatic residue is linearly amplified by the number of such mutations and screened by the square root of the length of the protein ($\sqrt{L}$), indicating a disruption in the percolated network of interactions sustaining the condensate. Condensates made of longer PLDs are more robust against aromatic deletions because, by construction, each aromatic residue represents a smaller percentage of the total valency of the PLD in question. In other words, longer PLDs are higher valency molecules than their shorter counterparts and can, thus, form a densely connected condensed-liquid network, even when a few of their aromatic stickers have been deleted.

Our findings also highlight the crucial role of Arg in the critical parameters of PLD condensates. The scaling law for Arg deletions ($640K/L$) predicts that the destabilizing effect of an Arg mutation to an uncharged, non-aromatic residue is inversely proportional to the length of the protein ($L$), as well as linear with respect to the number of mutations. Specifically, it indicates that, for a PLD of the length of 100 amino acids, the change in the critical temperature will be of the order of 6.4 K (see **Table 1**). While formulaically similar to the scaling law for aromatic deletions, a notable difference is a change in the dependence on the PLD length. In the case of Arg deletions, the effect of mutations is screened by $L$ rather than $\sqrt{L}$, indicating a stronger screening dependence on the PLD length. This indicates that for long PLDs, the destabilizing effects due to Arg deletions are diminished when compared to aromatic deletions, while in shorter PLDs this trend is reversed with each Arg deletion exerting a more significant destabilizing effect than the aromatic deletions.

These results underscore once again how for PLDs cation–$\pi$ and $\pi$–$\pi$ interactions are the main drivers of phase separation (**Vernon et al., 2018**; **Das et al., 2020**; **Bremer et al., 2022**; **Martin et al., 2020**; **Krainer et al., 2021**; **Fisher and Elbaum-Garfinkle, 2020**; **Fossat et al., 2021**; **Dyson et al., 2006**; **Andrew et al., 2002**). While it may seem counterintuitive for Arg deletions to have a lesser effect in the critical parameters of longer PLDs than their aromatic counterparts (as Arg residues are known to form strong cation–$\pi$ interactions **Park et al., 2023**; **Gallivan and Dougherty, 2000**), an explanation lies in the type of systems in question, which are particularly rich in aromatic residues. Longer PLDs have, as a result of their higher valency, a larger network of possible pairwise interactions, which can mitigate the destabilizing effect of Arg deletions. In the case of shorter PLDs, each Arg residue represents a larger proportion of the total valency, and thus its deletion has a proportionally larger impact on the critical parameters. In contrast, aromatic residues contribute to the stability of the condensate through both cation–$\pi$ and $\pi$–$\pi$ interactions, which are predicted to be weaker than their cation–$\pi$ analogues (**Krainer et al., 2021**; **Joseph et al., 2021**), but more numerous in PLDs. In a longer PLD, the deletion of an aromatic residue results in the loss of more potential interactions, creating a larger destabilizing effect than the deletion of an Arg residue. However, in shorter PLDs,

due to their lower total valency, the loss of potential associative interactions that result from the deletion of an aromatic residue is lower. Hence, the destabilizing effect resulting from this mutation is comparatively less than for the Arg deletion.

Our scaling laws have also allowed us to quantify the destabilizing effect of Arg to Lys mutations in the phase separation of PLDs. Concretely, mutations of Arg residues to Lys result in an overall decrease in the critical temperature of phase separation, twofold in comparison to the impact of mutating Arg into an alternate uncharged non-aromatic residue. Consequently, the scaling law considerably intensifies for mutations from Arg to Lys, nearly doubling the effects of each mutation in the critical temperature of the condensate (see *Table 1*). This relationship implies that each mutation of Arg to Lys exerts a substantially more destabilizing effect than mutations of Arg to other uncharged, non-aromatic residues, consistent with experimental evidence (*Bremer et al., 2022*; *Fisher and Elbaum-Garfinkle, 2020*; *Ukmar-Godec et al., 2019*; *Greig et al., 2020*) and highlighting the significant, modulatory role Lys plays in the phase behavior of PLDs.

Uncharged, non-aromatic residues such as Ser, Gly, Asn, Gln, and Thr are found to play a marginal role in PLD condensate stability (*Bremer et al., 2022*; *Martin and Mittag, 2018*; *Holehouse et al., 2021*; *Choi and Pappu, 2020c*). Several PLD condensates are stable in the face of Gly, Ser, Thr, and Ala mutations. This behavior is quantified by their respective scaling laws, with values of the scaling constants for mutations from Gly to Thr, Ser to Thr, and Ala to Ser at $-0.09\,\mathrm{K}$, $-0.08\,\mathrm{K}$, and $0.16\,\mathrm{K}$, respectively. These numerical values indicate that, for a change of one degree in the critical temperature, approximately 10 amino acids of this set would need to undergo mutation. Additionally, these scaling laws do not display a significant length dependency. This observation aligns with the idea that these mutations primarily affect the condensate properties through modulation in excluded volume and solvation, rather than by interrupting specific interaction networks.

In contrast, the condensates display heightened sensitivity towards Asn and Gln mutations, underscored by the Asn to Gln scaling of $50\,\mathrm{K}/L$. This law is also inversely length-dependent, indicating that mutations on these amino acids affect interaction networks and that Asn to Gln mutations result in an increased propensity of the system to phase separate. We hypothesize that, in addition to differences in solvation volume, the polarity of each amino acid plays a role by forming charge–dipole and dipole–dipole interactions. Both Gln and Asn are polar amino acids, with Gln having a longer side chains than Asn, which could potentially interact with a larger number of nearby atoms. Furthermore, the additional methylene group (-CH2-) may slightly alter the electron distribution around the amide group, which can lead to the formation of more stable dipole-dipole contacts. It is important to note that our computational model, Mpipi, is a charge-fixed model, and hence does not directly capture the effects of polarization. However, the lack of physical details in the Mpipi model is compensated by the parameters having been derived from bioinformatics data.

While the Mpipi model has proven to be a powerful tool in predicting the phase behaviour of PLDs, it is important to acknowledge its limitations. For example, the model does not account for temperature-dependent solvent effects, such as variations in hydrophobic interactions and charge-mediated interactions facilitated by desolvation penalties. These deficiencies imply that the model might not accurately predict the effects of mutations in particular sequences, such as sequences high in proline or aliphatic residues, where the interaction landscape is temperature sensitive. Additionally, the Mpipi model is not predictive of folding, or secondary structure. An assumption in this work is that the mutations introduced do not significantly alter the secondary structure of each PLD, which is a necessary simplification within the scope of this study.

Collectively, the definition of a set of scaling laws acts as a unifying framework for the study of the critical parameters of PLD phase separation, by allowing for direct comparison of individual sequence effects. Such laws are cumulative, meaning that the effects of successive mutations can be predicted by the addition of the appropriate laws, as long as the resulting mutation keeps within PLD composition and patterning biases.

Our work explores a new framework to describe, quantitatively, the phase behavior of a large set of protein variants via the identification of scaling laws. Our scaling laws can predict changes in the stability of condensates based on amino acid sequence mutations. This code not only allows for testing the effects of composition bias on PLD phase separation but also serves as a predictive tool, allowing for a better selection of mutations to achieve desired changes in different condensates. This study is focused on PLDs, but we hypothesize similar scaling behavior in other IDPs, with the laws

having a dependence on the protein length that reflects the nature of the underlying interaction networks of the system. Our findings have important implications for understanding PLD condensation and for designing modifications to counter their aberrant aggregation, by setting a basis for characterizing and quantifying the molecular determinants of PLD phase behavior.

## Methods

### Design of the dataset

To design sequence variants to test the phase behavior of prion-like domains, we use in-silico targeted mutagenesis to introduce specific changes to the amino acid sequence of the PLD of interest. These changes involve substituting different amino acids, while carefully maintaining patterning and compositional biases, to test the role of given amino acids in the phase-separating propensities of the respective PLDs.

The nomenclature employed is consistent with that of Bremer and colleagues (*Bremer et al., 2022*). Namely, if $n$ amino acids of type $X$ are removed from the sequence and replaced with $n$ amino acids of type $Y$, the variant is labeled $-nX + nY$. If instead, $n$ amino acids of type $X$ are removed from the sequence and replaced with amino acids (often different types) in proportion to the compositional bias of the PLD, the variant is simply labeled $-nX$.

To begin, we select a family of PLDs composed of hNRNPA1, RBM14, FUS, TDP43, TIA1, and EWSR1 and perform mutations on each sequence to generate a set of variants. Accordingly, we assess whether the effects of sequence mutations in PLD phase behaviors hold for the whole family of PLDs. Furthermore, our data set allows us to develop a set of rules that can aid in understanding and predicting PLD phase behavior. The amino acid sequences of the PLDs, and their corresponding variants, investigated herein are provided in Appendix 1.

In total, we compute phase diagrams of 140 PLD variants via MD. This task required a large amount of CPU hours, over a 1.25 million in total. Given these demands, efficient utilization of high-performance computing (HPC) resources and proper parallelization of the computational tasks are paramount to ensuring optimal performance. To this end, we compute MD trajectories using the LAMMPS (Large-scale Atomic/Molecular Massively Parallel Simulator) software (*Thompson et al., 2022*), exploiting its efficient parallelization algorithms. Another significant computational challenge involves the large size of data generated. Each single run produced a significant amount of trajectory, backup, and analysis data, and a large number of files. On average, each run produced over 50 files and a minimum of 5 GB of data. To efficiently handle this, we implement a robust, tiered, data storage and retrieval system, ensuring that the vast amounts of data are organized through a combination of indexing and partitioning, enhancing search efficiency and accessibility. We leveraged the Lustre filesystem for its high performance and scalability, making it the ideal choice for managing and accessing large volumes of active data required by our computational jobs. Custom scripts are employed for automated data processing, enabling us to extract meaningful insights from the complex datasets. All scripts are available in our GitHub repository (copy archived at *Maristany, 2024*).

### Critical parameter determination through MD

We used MD simulations to determine the full binodal (i.e. the boundaries that delimit the two-phase region from the one-phase region) for the PLDs. Specifically, we perform MD simulations in the canonical ensemble—i.e., fixed volume ($V$), temperature ($T$), and number of particles ($N$). For each binodal, a series of independent $NVT$ simulations are performed at different temperatures and the density of each of the coexisting phases are determined. Thus, this approach yields binodal in the temperature–density phase space.

#### Direct coexistence simulations

Specifically, we use Direct Coexistence simulations, which enables the simultaneous simulation of both high- and low-density phases of a mixture, separated by an interface, within a single simulation box. Additionally, these simulations are performed with periodic boundary conditions in each dimension.

To initiate the $NVT$ Direct Coexistence simulations, we start with one copy of a PLD in a cubic simulation cell and replicate it to the target number of chains (64 in total). We then iteratively scaled the simulation box to compress the chains into a high-density slab (approx. 0.8–1 g/cm³). Here, we

use a Langevin thermostat (*Grønbech-Jensen and Farago, 2013*) (at the target temperature, with a damping time of 40 ps) for temperature control. Keeping the length of the box in the X and Y dimensions fixed, we then elongate the Z dimension of the box. We then simulate each system at the target temperature for over 200 ns (see Convergence test in Appendix 1). From each simulation, we compute the density profile in the Z direction and determine the density of the dilute and condensed phases. These data are then used to produce the binodals.

## Mpipi model

All the simulations are run using the Mpipi coarse-grained force-field (*Joseph et al., 2021*) with an implicit solvent. In the Mpipi model electrostatic, cation–$\pi$ and $\pi$–$\pi$ interactions are carefully balanced to capture the phase behavior of PLDs. Mpipi achieves quantitative agreement with experiments in terms of prediction of critical solution temperatures of such systems (*Joseph et al., 2021*). In Mpipi, each amino acid is represented by a single unique bead—i.e., of corresponding mass, molecular diameter ($\sigma$), charge ($q$), and an energy scale that reflects the relative frequency of planar $\pi$–$\pi$ contacts ($\epsilon$) (*Joseph et al., 2021*).

The potential energy in the Mpipi model is computed as follows:

$$E_{\mathrm{mpipi}} = E_{\mathrm{bond}} + E_{\mathrm{elec}} + E_{\mathrm{pair}}. \tag{16}$$

Beads are connected via harmonic springs

$$E_{\mathrm{bond}} = \sum_i \frac{1}{2} k (r_i - r_i^0)^2. \tag{17}$$

Here $k$ is the bond force constant (19.1 kcal mol$^{-1}$ Å$^{-2}$) and $r_i^0$ is the equilibrium bond length, which is set to 3.81 Å.

The electrostatic contribution is modeled via a Coulomb interaction term, with Debye–Hückel screening.

$$E_{\mathrm{elec}} = \sum_{i,j} \frac{q_i q_j}{4\pi \epsilon_r \epsilon_0 r_{ij}} \exp(-\kappa r_{ij}), \tag{18}$$

where $q_i$ and $q_j$ are the charges of the bead $i$ and $j$, respectively. $\epsilon_0$ is the electric constant and $\epsilon_r = 80$ is the relative dielectric constant of water. $\kappa = 0.126$ Å$^{-1}$ is the inverse Debye screening length, which corresponds to a monovalent salt concentration of 150 mM.

Finally, short-ranged non-bonded interactions are modeled with the Wang–Frenkel potential (*Wang et al., 2020*)

$$E_{\mathrm{pair}} = \sum_{i,j} \phi_{i,j} = \sum_{i,j} \epsilon_{i,j} \alpha_{i,j} \left[ \left( \frac{\sigma_{i,j}}{r} \right)^{2\mu_{i,j}} - 1 \right] \left[ \left( \frac{\sigma_{i,j}}{r} \right)^{2\mu_{i,j}} - 1 \right]^{2\nu_{i,j}} \tag{19}$$

where $E_{\mathrm{pair}}$ is the potential energy between all pairs of amino acids, $\phi_{i,j}$ is the potential energy between the pairs of amino acids $i$ and $j$, $\epsilon_{ij}$ is an energy parameter that dictates the strength of interaction of that particular pair, $\sigma_{i,j}$ is average the Van der Waals radii of the pair, $r$ is the distance between center of masses and $\nu_{i,j}$ and $\mu_{i,j}$ are fitted exponential parameters that determine the shape of the energy curve. Full details of the Mpipi model are described in *Joseph et al., 2021*.

## Data fitting

To estimate the critical temperatures of the binodals, we use the law of coexistence densities

$$\left( \rho_{\mathrm{high}}(T) - \rho_{\mathrm{low}}(T) \right)^{3.06} = d \left( 1 - \frac{T}{T_c} \right) \tag{20}$$

The constant 3.06 in the equation is a dimensionless empirical factor that was derived from simulations of the 3D Ising model. The critical densities are computed by assuming that the law of rectilinear diameters holds, namely

$$\rho_{\text{high}}(T) + \rho_{\text{low}}(T) = 2\rho_c + 2A(T - T_c) \tag{21}$$

where $\rho_{\text{high}}(T)$, $\rho_{\text{low}}(T)$ and $\rho_c$ are the densities of the high-density and low-density phases and the critical density, respectively, $T_c$ is the critical temperature, and $d$ and $A$ are fitting parameters.

## Finite-size scaling

In direct coexistence simulations in the slab geometry, finite-size effects can arise from either the cross-sectional area (i.e. size of the interface) or from the long axis (i.e. fraction bulk condensate). To ensure our calculations are robust with respect to system size, we perform finite-size scaling at fixed system number density: (1) varying the size of the cross-section and (2) varying the length of the long axis. Using these tests, we confirm that the phase diagrams computed for systems of varying sizes (27, 64, 125 and 216 chains) exhibit no significant difference in predicted densities (see *Appendix 1—figure 8*). Our scaling analysis revealed that 64 chains in a box of dimensions 12.5×12.5×100 nm is the minimum size that ensures proteins do not interact with their periodic images and that the condensed phase is large enough to represent a true bulk phase.

## Acknowledgements

We thank Pin Yu Chew for the useful comments on the manuscript. This project has received funding from the European Research Council (ERC) under the European Union's Horizon 2020 research and innovation program (grant agreement No 803326 to RCG). MJM acknowledges the Winton Programme for Physics of Sustainability for doctoral funding. AAG is funded by the ERC (grant agreement No 803326). JRE acknowledges funding from the Ramon y Cajal fellowship (RYC2021-030937-I) and the Spanish National Agency for Research under the grant PID2022-136919NA-C33. This work was supported by the Engineering and Physical Sciences Research Council (EPSRC) [grant number EP/X02332X/1 to JH] under the UK Research and Innovation (UKRI) Postdoctoral Fellowships Guarantee Scheme [project TF-CHROM-LLPS]. JAJ acknowledges research support from departmental start-up funds provided by the Department of Chemical and Biological Engineering and the Omenn–Darling Bioengineering Institute at Princeton University. JAJ also acknowledges research support from the Chan Zuckerberg Initiative DAF (an advised fund of Silicon Valley Community Foundation; grant 2023–332391) and the National Institute of General Medical Sciences of the National Institutes of Health under Award Number R35GM155259. This project made use of time on HPC granted via the UK High-End Computing Consortium for Biomolecular Simulation, HECBioSim (http://hecbiosim.ac.uk), supported by EPSRC (grant no. EP/R029407/1).

## Additional information

### Competing interests

Rosana Collepardo-Guevara: Reviewing editor, eLife. The other authors declare that no competing interests exist.

### Funding

| Funder | Grant reference number | Author |
|---|---|---|
| European Research Council | 803326 | Anne Aguirre Gonzalez Rosana Collepardo |
| Ramon y Cajal Fellowship | RYC2021-030937-I | Jorge R Espinosa |
| Winton Programme for Physics of Sustainability | | M Julia Maristany |
| Spanish National Agency for Research | PID2022-136919NA-C33 | Jorge R Espinosa |
| Engineering and Physical Sciences Research Council | EP/X02332X/1 | Jan Huertas |

| Funder | Grant reference number | Author |
|---|---|---|
| Department of Chemical and Biological Engineering and the Omenn--Darling Bioengineering Institute at Princeton University | | Jerelle A Joseph |
| Chan Zuckerberg Initiative | 2023-332391 | Jerelle A Joseph |
| National Institutes of Health | R35GM155259 | Jerelle A Joseph |

The funders had no role in study design, data collection and interpretation, or the decision to submit the work for publication.

### Author contributions

M Julia Maristany, Conceptualization, Data curation, Formal analysis, Validation, Investigation, Visualization, Methodology, Writing – original draft, Writing – review and editing; Anne Aguirre Gonzalez, Formal analysis, Writing – review and editing; Jorge R Espinosa, Jan Huertas, Writing – review and editing; Rosana Collepardo-Guevara, Conceptualization, Resources, Supervision, Funding acquisition, Investigation, Visualization, Methodology, Writing – original draft, Writing – review and editing; Jerelle A Joseph, Conceptualization, Supervision, Funding acquisition, Investigation, Visualization, Methodology, Writing – original draft, Writing – review and editing

### Author ORCIDs

M Julia Maristany ⓘ https://orcid.org/0009-0009-8875-9225
Jorge R Espinosa ⓘ https://orcid.org/0000-0001-9530-2658
Rosana Collepardo-Guevara ⓘ https://orcid.org/0000-0003-1781-7351
Jerelle A Joseph ⓘ https://orcid.org/0000-0003-4525-180X

Reviewer #1 (Public review): https://doi.org/10.7554/eLife.99068.3.sa1
Reviewer #2 (Public review): https://doi.org/10.7554/eLife.99068.3.sa2
Reviewer #3 (Public review): https://doi.org/10.7554/eLife.99068.3.sa3
Author response https://doi.org/10.7554/eLife.99068.3.sa4

## Additional files

### Supplementary files

MDAR checklist

### Data availability

The source code for running the Mpipi model in LAMMPS, example input scripts, as well as instructions for reproducing the data through a demo can be found in the figshare data repository at https://doi.org/10.6084/m9.figshare.28198910. The data supporting the findings of this manuscript are also available in the figshare data repository.

The following dataset was generated:

| Author(s) | Year | Dataset title | Dataset URL | Database and Identifier |
|---|---|---|---|---|
| Maristany MJ, Aguirre A, Espinosa J, Huertas J, Collepardo-Guevara R, Joseph J | 2025 | Code and Source Data for "Decoding Phase Separation of Prion-Like Domains through Data-Driven Scaling Laws" | https://doi.org/10.6084/m9.figshare.28198910.v1 | figshare, 10.6084/m9.figshare.28198910.v1 |

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

## Appendix 1

### Full amino acid sequences
RBM14

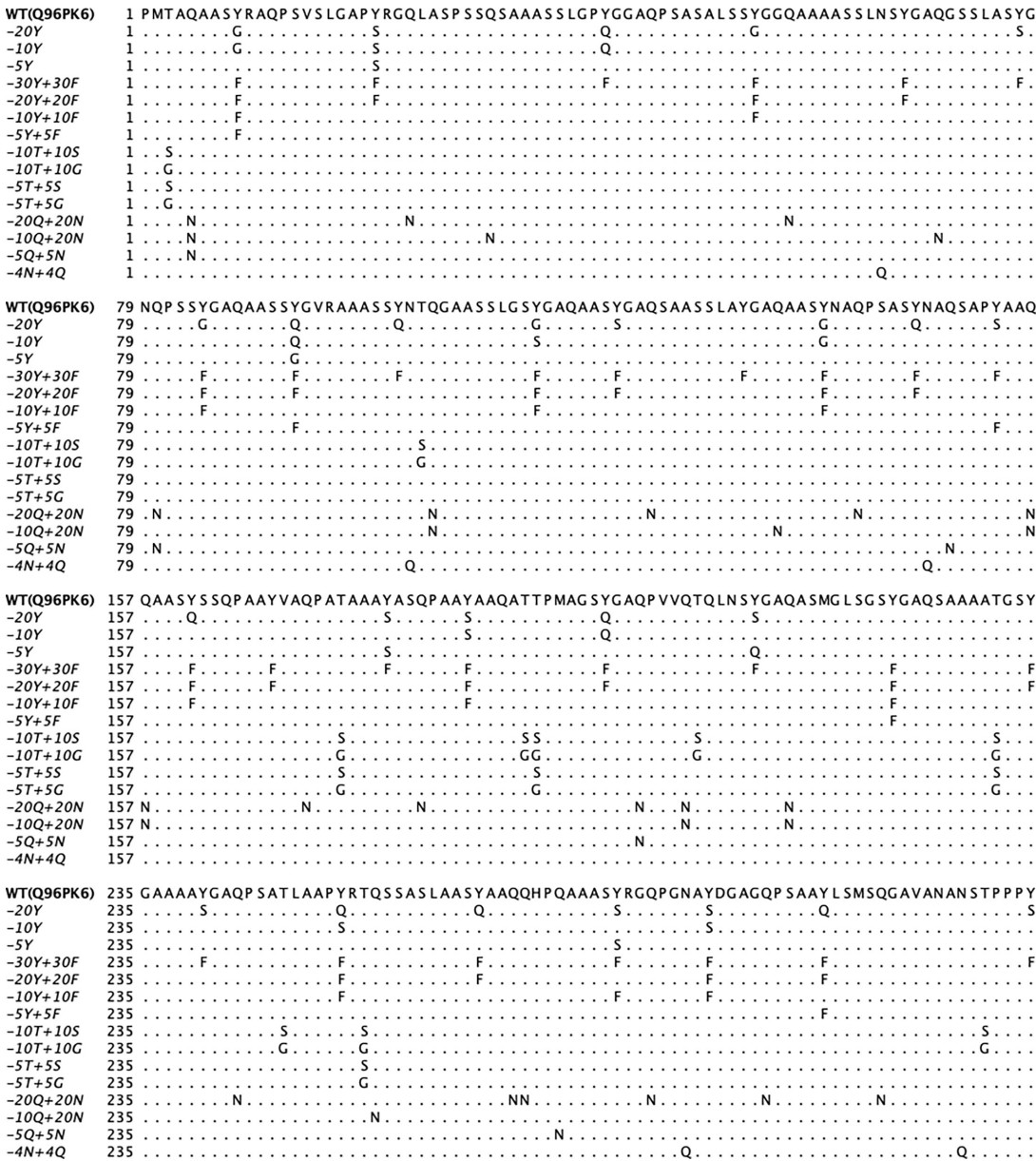

**Appendix 1—figure 1.** Designed variants of RBM14 prion-like low complexity domain (PLD).

## TDP43

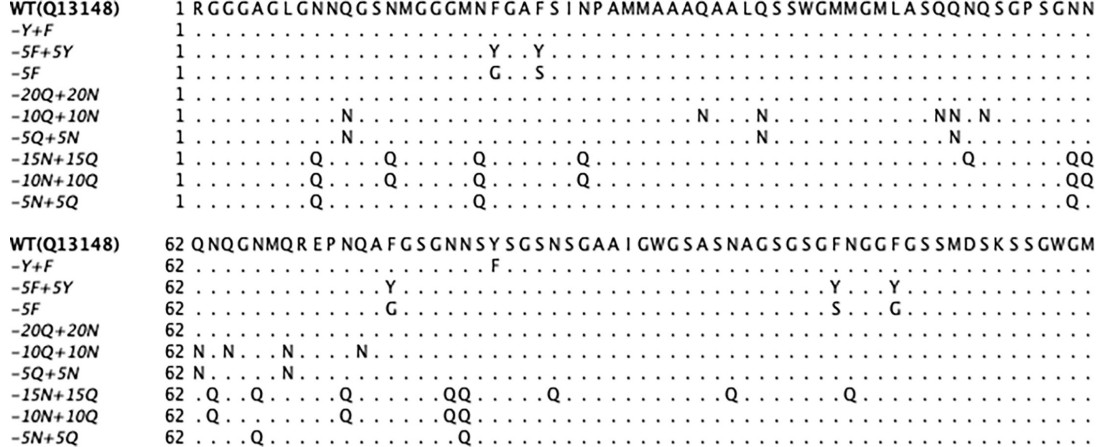

**Appendix 1—figure 2.** Designed variants of TDP43 prion-like low complexity domain (PLD).

## TIA1

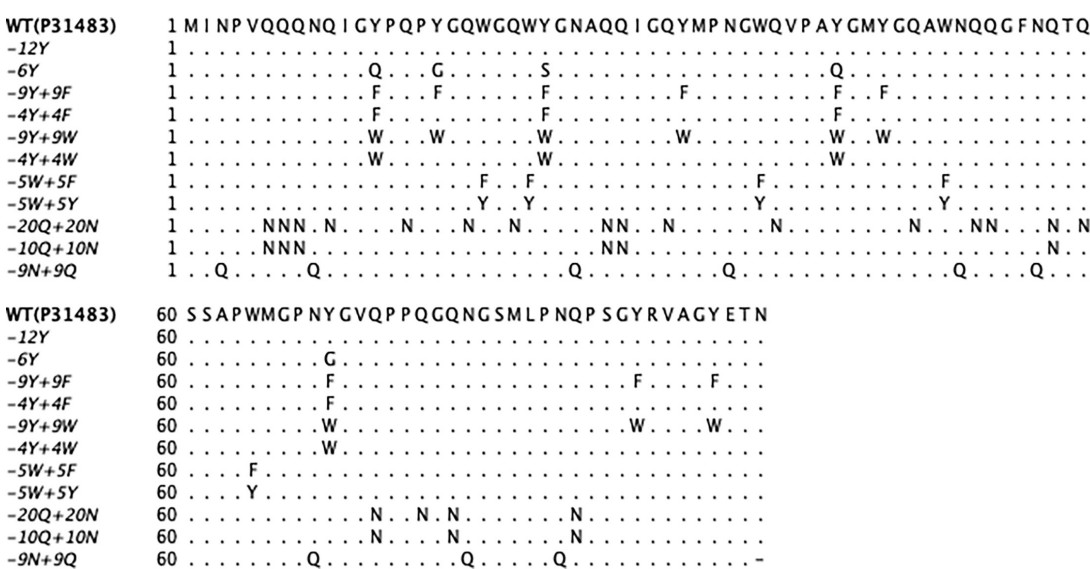

**Appendix 1—figure 3.** Designed variants of TIA1 prion-like low complexity domain (PLD).

FUS

```
(P35637)      1 MASNDYTQQATQSYGAYPTQPGQGYSQQSSQPYGQQSYSGYSQSTDTSGYGQSSYSSYGQS
-28Y+28F      1 .....F........F..F.......F......F....F..F........F....F..F...
-14Y+14F      1 .....F..........F.........F.......F.......F..............F...
-7Y+7F        1 .....F..........F.........F.......F..........F..............
-14Y          1 ................Q..........G..........S..........Q.......G...
-7Y           1 ...................Q..................G.....................
-10T+10S      1 .....S...S.......S.......................S.S................
-10T+10G      1 .....G...G.......G.......................G.G................
-5T+5S        1 .....S...........S......................S...................
-5T+5G        1 .....G...........G......................G...................
-10G+10T      1 ...............T......................T........ ............
-10S+10T      1 ..T................................T.............T...
-20Q+20N      1 .......N...N.........N....N.......N......N............N.
-10Q+10N      1 .......N............N.......N....................N.
-5Q+5N        1 .......N..................N.
-4N+4Q        1

(P35637)     62 QNTGYGTQSTPQGYGSTGGYGSSQSSQSSYGQQSSYPGYGQQPAPSSTSGSYGSSSQSSSY
-28Y+28F     62 ....F........F.....F.......F.....F..F........F.........F
-14Y+14F     62 ....F............F............F.........F..........
-7Y+7F       62 ................F.............F........
-14Y         62 ...............S..........Q..........G.................S
-7Y          62 ...............S..................Q.................
-10T+10S     62 ..S...S..S......S................S..............
-10T+10G     62 ..G...G..G......G................G..............
-5T+5S       62 ......S.........S................S..............
-5T+5G       62 ......G.........G...............................
-10G+10T     62 ................T.....................T.......
-10S+10T     62 ...............T..............T.........T..T..
-20Q+20N     62 ......N.......N......N.......N.........N....
-10Q+10N     62 ..............N.......N........N........
-5Q+5N       62 ..................N.........
-4N+4Q       62 .Q.......................

(P35637)    123 GQPQSGSYSQQPSYGGQQQSYGQQQSYNPPQGYGQQNQYNSSSGGGGGGGGGGGGNYGQDQSS
-28Y+28F    123 .......F....F.....F.....F......F......F...........F.....
-14Y+14F    123 .......F..........F........F..............F....
-7Y+7F      123 .................F............F....
-14Y        123 ..............Q.........S........G...............
-7Y         123 ..............S.................G...............
-10T+10S    123 ...............................................
-10T+10G    123 ...............................................
-5T+5S      123 ...............................................
-5T+5G      123 ...............................................
-10G+10T    123 .................T.......T.......T.......
-10S+10T    123 ...........T................T...........
-20Q+20N    123 ...N......N.....N....N.N........N..N............N..
-10Q+10N    123 ...N..........N....N.......N..........
-5Q+5N      123 .................N...............N..
-4N+4Q      123 .............................Q..Q...........Q......

(P35637)    184 MSSGGGSGGGYGNQDQSGGGGSGGYGQQDRGGRGRGGSGGGGGGGGGGGGYNRSSG
-28Y+28F    184 .............F............F.................F.....
-14Y+14F    184 .............................F................
-7Y+7F      184 ...........................................
-14Y        184 .........G..........................G.....
-7Y         184 .........G.................................
-10T+10S    184 ...........................................
-10T+10G    184 ...........................................
-5T+5S      184 ...........................................
-5T+5G      184 ...........................................
-10G+10T    184 .......T.........T.........T.......
-10S+10T    184 ..T........................................
-20Q+20N    184 ...........................................
-10Q+10N    184 ...........................................
-5Q+5N      184 ...........................................
-4N+4Q      184 ..........Q................................
```

**Appendix 1—figure 4.** Designed variants of Fused in Sarcoma (FUS) prion-like low complexity domain (PLD).

EWSR1

```
WT(Q01844)      1 MASTDYSTYSQAAAQQGYSAYTAQPTQGYAQTTQAYGQQSYGTYGQPTDVSYTQAQTTATYGQTAYATSYGQP
-20Y            1 .....T..Q........S..G.................T.........Q.............S......
-10Y            1 .....T..Q........S..G.................................T.....
-5Y             1 .....T...........................................................Q...
-30Y+30F        1 .....F..F..........F.......F......F.....F.......F.......F...F...F...
-20Y+20F        1 .....F...........F..........F......F.......F.......F......F.......
-10Y+10F        1 .....F...........F..........F......F.......F......
-5Y+5F          1 .....F...............................F.....
-30T+30S        1 ...S..........S...S......S........S...S....S....S.S...S......
-30T+30G        1 ...G..........G...G......G........G...G...G...G.G...G......
-20T+20S        1 ...S..........S.......S.......S........S....S....S.....
-10T+10S        1 ......S..................S...............S......S......
-20T+20G        1 ...G..............G......G........G.......G...G...G......
-10T+10G        1 ...G..............G......G.............G...G...G......
-30S+30T        1 ..T...T..T.............T.........T......T.........T....
-20G+20T        1 ................T........T.......T..T...........T..
-10G+10T        1 ................T................T...........T..
-20S+20T        1 ..T.....T.............T.........T...............T....
-10S+10T        1 ..T..............T..............T...........T....
-20Q+20N        1 .........N....N...........N......N......N......N......N.
-10Q+10N        1 .........N..............N.........N.........
-5Q+5N          1 .........N......................N.........
-4N+4Q          1

WT(Q01844)     74 PTGYTTPTAPQAYSQPVQGYGTGAYDTTTATVTTTQASYAAQSAYGTQPAYPAYGQQPAATAPTRPQDGNKPT
-20Y           74 ...T........G..........Q..................T..S.....
-10Y           74 ..Q.............S................G.....
-5Y            74 ...............S................G.....
-30Y+30F       74 ...........F.....F....F.........F.....F.......F...........
-20Y+20F       74 ...F...........F....F..........F.....F.......F...
-10Y+10F       74 ..........F................F.....F...
-5Y+5F         74 ...................F.....
-30T+30S       74 .S...SS..S..............SSS...SSS.............S.......S
-30T+30G       74 ....GG..G.............GG..G.GG...........G..........G........G
-20T+20S       74 .S...S..........S.....S..S..S..........S.......S......
-10T+10S       74 ......S.............S....S...............S
-20T+20G       74 .G...G...........G....G..G..G..........G.........G......
-10T+10G       74 ......G.............G....G..........G......
-30S+30T       74 ................T....T...........
-20G+20T       74 ..T..............T..T............T.........T.......T....
-10G+10T       74 ................T............T.........T.......T....
-20S+20T       74 ..................T.........
-10S+10T       74 ...........................
-20Q+20N       74 ...........N...........N..........N......N.....
-10Q+10N       74 ...........N..................N.....
-5Q+5N         74 ..........................N.....
-4N+4Q         74 ...............................Q...

WT(Q01844)    147 ETSQPQSSTGGYNQPSLGYGQSNYSYPQVPGSYPMQPVTAPPSYPPTSYSSTQPTSYDQSSYSQQNTYGQPSS
-20Y          147 ....................Q.G......T.....................S...........Q.....
-10Y          147 ...........T...............................S.........
-5Y           147 ..............T...................................
-30Y+30F      147 ..........F.....F....F.........F.....F.......F.....F.....F.....
-20Y+20F      147 ...........F.....F.........F.......F.......F.....F.....
-10Y+10F      147 ...............F..............F.......F.....
-5Y+5F        147 .................F...........
-30T+30S      147 .S...............................S.......S....S..S.........S......
-30T+30G      147 .G......G...................G.....G...G..G.........G.....
-20T+20S      147 .S...........................S.......S.........S.....
-10T+10S      147 ...................................S.........
-20T+20G      147 .G...........................G.........G.........G.....
-10T+10G      147 ..............................G.........G....
-30S+30T      147 ..T....T........T....T.......T.......T...T....T...TT.T........TT
-20G+20T      147 ...........T......T.T........T..................
-10G+10T      147 ...........T.............T........
-20S+20T      147 ..T....T........T.......T.....T.T........T..T........T
-10S+10T      147 ..T.............T.......................T..............T.
-20Q+20N      147 .....N..............N...........N.........N.....
-10Q+10N      147 ..............N..............N.....
-5Q+5N        147 ..............................N.....
-4N+4Q        147 ...........Q........Q...............................Q......

WT(Q01844)    220 YGQQSSYGQQSSYGQQPPTSYPPQTGSYSQAPSQYSQQSSSYGQQSSFRQDHPSSMGVYGQ
-20Y          220 T....................G...............S.............
-10Y          220 ..................................................
-5Y           220 ..................................................
-30Y+30F      220 F.....F.....F.......F.....F...........
-20Y+20F      220 ......F.........F...........F...........F..
-10Y+10F      220 F.................................................
-5Y+5F        220 ..................................................
-30T+30S      220 ..............S.....S......................
-30T+30G      220 ..............G.....G......................
-20T+20S      220 ..................S......................
-10T+10S      220 ..............S......................
-20T+20G      220 ..................G......................
-10T+10G      220 ..................................................
-30S+30T      220 .....T....TT............T.....T..T...T....T.....
-20G+20T      220 .T.....T..............T...........T.........T..T.
-10G+10T      220 .......T..............T...........T....
-20S+20T      220 .....T....T........T....T...T.T..........
-10S+10T      220 ..........T...........T.........T....
-20Q+20N      220 ..N.....N.....N..........N.....N.........
-10Q+10N      220 ..N.................N.............N.....
-5Q+5N        220 ..................N......................
-4N+4Q        220 ..................................................
```

**Appendix 1—figure 5.** Designed variants of EWSR1 prion-like low complexity domain (PLD).

Dataset representation

## Convergence tests and finite-size scaling

Molecular Dynamics simulations are inherently sensitive to both the length of simulated time and system scaling factors such as the number of particles present in the system and the size of the simulation box. In this study, we systematically tested all these aspects to ensure proper convergence of the results.

The system was placed in the middle of the simulation box, and the length of X and Y was the same as the length of the condensate, while the box was elongated in Z, to about ten times the size of X, and Y (see 1). Importantly, the cross-section is large enough to avoid protein self-interactions through the periodic boundary conditions (e.g. cross-section sizes are at least twice the protein radius of gyration in condensate bulk conditions).

We tested for time convergence by plotting the obtained density of the protein wild types after increasing simulation times, from 50 to 350 ns. On simulated timesteps above 100 ns, the density of the condensate exhibited no significant differences. We thus concluded that, to optimize speed-up, 200 ns provided sufficient sampling for each point of the binodal.

Finally, we decided to keep the number of particles at a constant value of 64, which we found to be a good balance between minimizing finite-size effects and optimizing simulation speed (see *Appendix 1—figure 8*). We tested this by varying the corresponding number of protein chains of the FUS WT as a model protein from 27 to 216 and found that in these systems there was not a significant difference in the simulated density. Additionally, with 64 protein chains, we could ensure a bulk large enough to sustain the condition that the cross-section of the box should be higher than twice the radii of the gyration of the protein chain.

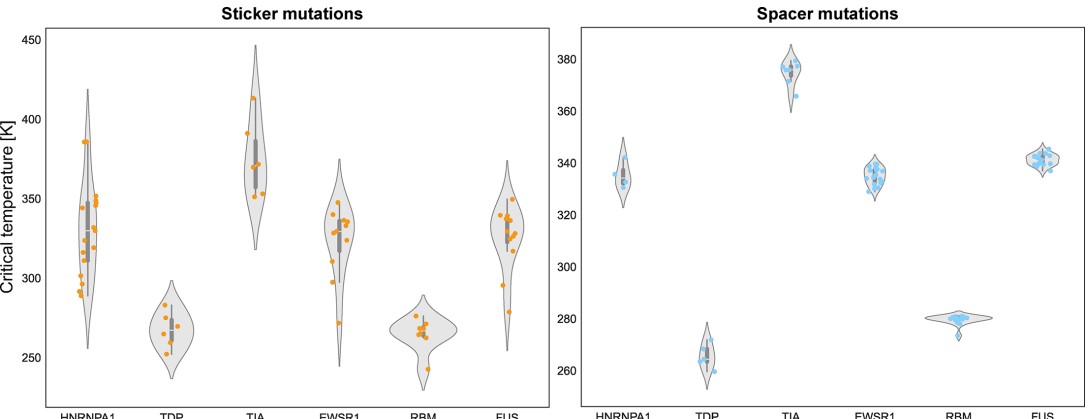

**Appendix 1—figure 6.** Representation of the entire computational data set, divided into mutations on 'sticker' or 'spacer' residues. Orange data points, on the left, represent variants where charged or aromatic residues were mutated (i.e. stickers), while cyan data points, on the right, represent all other types of mutations studied (i.e. spacers).

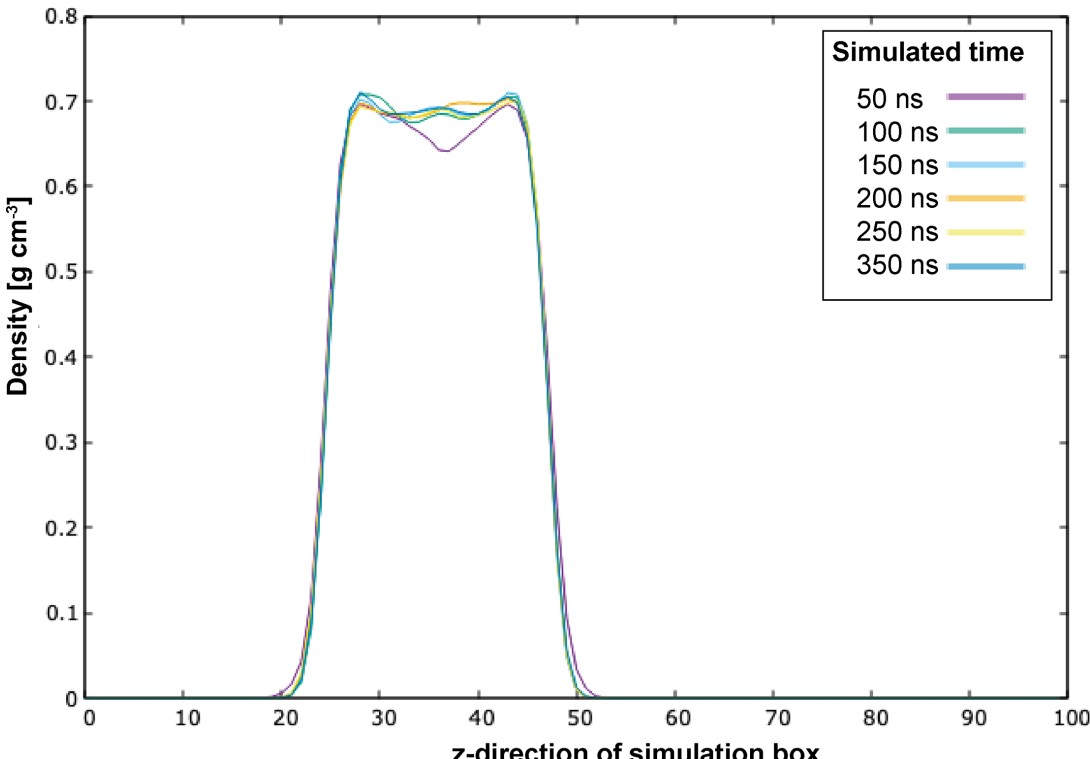

**Appendix 1—figure 7.** Convergence tests for the wild-type (WT) sequence of the prion-like low complexity domain (PLD) of Fused in Sarcoma (FUS). The density profiles of different and independent simulations are plotted across the perpendicular axis to the condensate interfaces. Each simulation was performed for different timescales (as specified in the legend) to check for proper convergence.

## Normalized and comparative contact maps

Further study of the differences between variants at the chemical level was carried out by computing the inter-molecular contacts per residue inside a phase-separated droplet. For each variant, the contact matrix was computed at a temperature around ten percent below the $T_c$. The cutoff to define a pairwise distance as a contact was dynamically calculated for each pair, as 1.2 times the average Van der Waals radius of both pair residues. Given residues $i$ and $j$ belong to different molecules, the contact maps were computed as shown below:

$$c_{i,j} = \sum_{i,j} \chi_{ij} \tag{22}$$

$$\chi_{ij} = \begin{cases} 1 & \text{if } |r_i - r_j| \leq r_{ij}^{\text{cutoff}} \\ 0 & \text{otherwise} \end{cases} \tag{23}$$

To get a more accurate picture of the main interactions driving phase separation for each variant, we normalized the contact maps to account for the occurrence of each residue in the sequence. This allowed us to account for the disparities in residue distributions in the sequences, and extract the relevant residue–residue interactions. For each residue-residue pair, the normalization was performed as follows:

$$C = \begin{vmatrix} \alpha_{M,M} & \alpha_{M,G} & \dots & \alpha_{M,I} \\ \alpha_{G,M} & \alpha_{G,G} & \dots & \alpha_{G,I} \\ \dots & \dots & \dots & \dots \\ \alpha_{I,M} & \alpha_{I,G} & \dots & \alpha_{I,I} \end{vmatrix} \tag{24}$$

$$\alpha_{i,j} = \frac{c_{i,j}}{n_i \cdot n_j} \tag{25}$$

where $n_i$ is the number of $i$ residues in the IDP sequence. The resulting contact matrix is min-max normalized:

$$\beta_{i,j} = \frac{\alpha_{i,j} - \min C}{\max C - \min C} \tag{26}$$

$$C_{\text{norm}} = \begin{vmatrix} \beta_{M,M} & \beta_{M,G} & \cdots & \beta_{M,I} \\ \beta_{G,M} & \beta_{G,G} & \cdots & \beta_{G,I} \\ \cdots & \cdots & \cdots & \cdots \\ \beta_{I,M} & \beta_{I,G} & \cdots & \beta_{I,I} \end{vmatrix} \tag{27}$$

The comparative contact maps (*Appendix 1—figure 9* and *Appendix 1—figure 10*), on the other hand, are computed as the difference in contacts of a protein variant respective to its wild-type, as:

$$C_{\text{comparative}} = C_{\text{variant}} - C_{\text{WT}} \tag{28}$$

which is finally mean-normalized.

$$\gamma_{i,j} = \frac{\alpha_{i,j} - \overline{C}_{\text{comparative}}}{\max C_{\text{comparative}} - \min C_{\text{comparative}}} \tag{29}$$

$$C_{\text{comparative,normed}} = \begin{vmatrix} \gamma_{M,M} & \gamma_{M,G} & \cdots & \gamma_{M,I} \\ \gamma_{G,M} & \gamma_{G,G} & \cdots & \gamma_{G,I} \\ \cdots & \cdots & \cdots & \cdots \\ \gamma_{I,M} & \gamma_{I,G} & \cdots & \gamma_{I,I} \end{vmatrix} \tag{30}$$

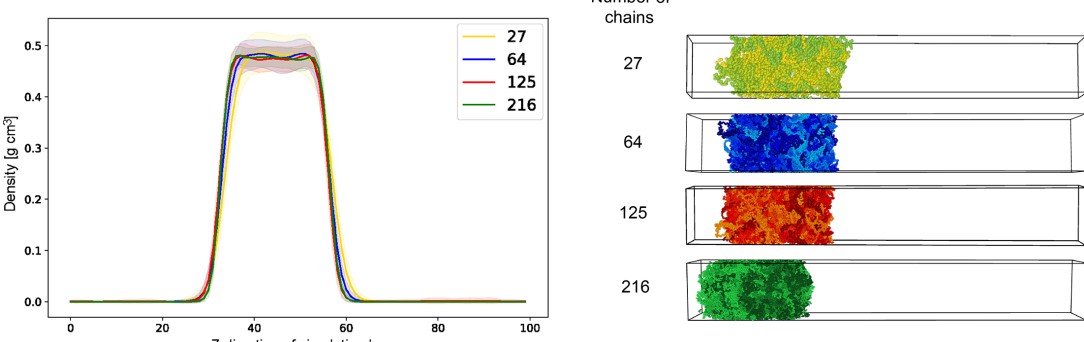

**Appendix 1—figure 8.** Finite-size scaling for the wild-type (WT) sequence of the prion-like low complexity domain (PLD) of Fused in Sarcoma (FUS). The density profiles of different and independent simulations are plotted across the perpendicular axis to the condensate interfaces. Each simulation was performed for different system sizes (aka, number of chains and corresponding box size, as specified in the legend) to check for proper convergence. To the right, a representation of the systems of increasing numbers of chains we employed to check for convergence. In each case, the overall density of the box is kept constant, while the cross-section and length are varied in order to maintain a constant density. For 27 chains, the box dimensions are 109×109×765 Å; for 64 chains 145×145×1021 Å; for 125 chains 182×182×1276 Å and for the final case of 216×218×218×1531 Å.

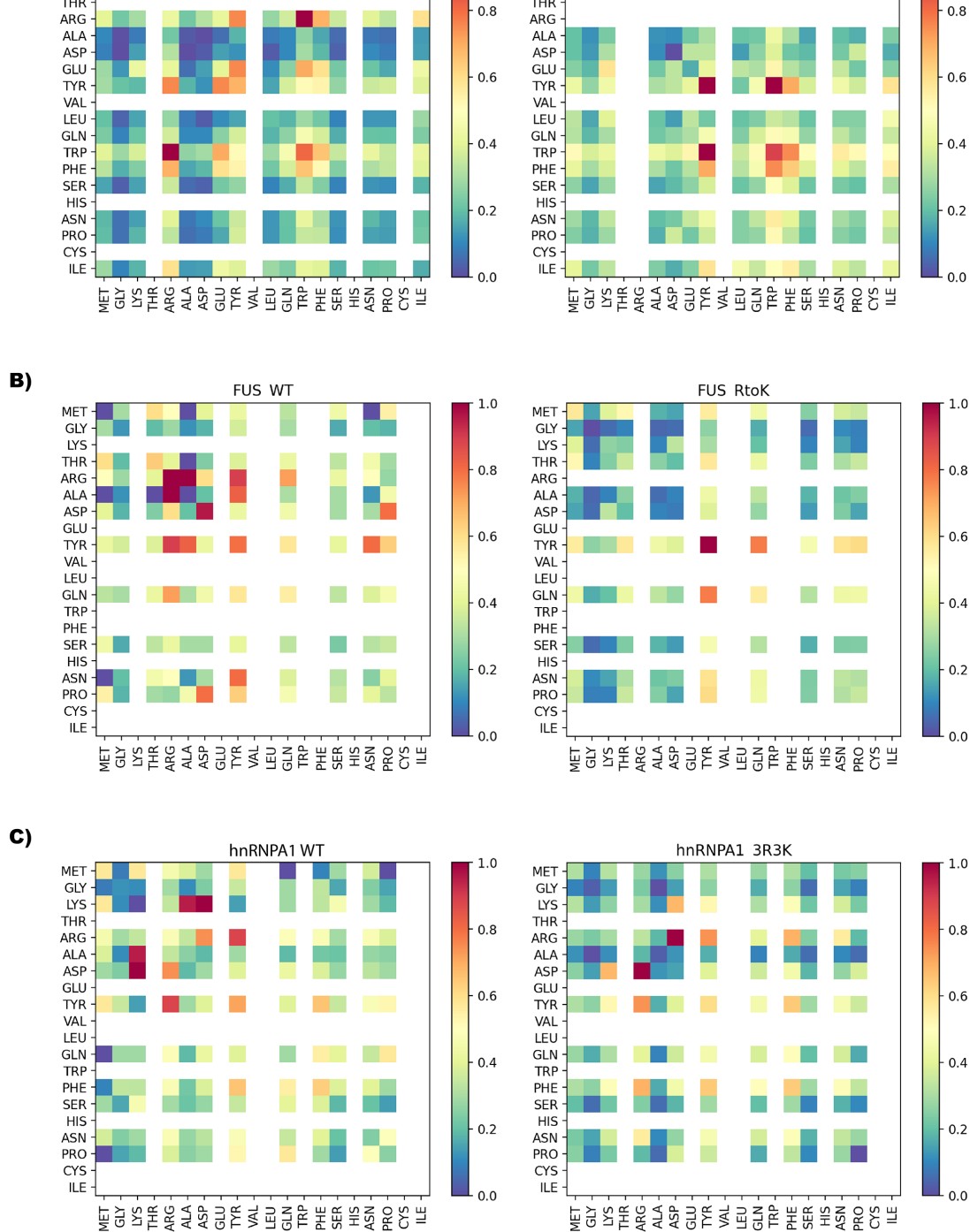

**Appendix 1—figure 9.** Normalized contact maps of different R to K variants of prion-like low complexity domains (PLDs): (**A**) TDP43 (**B**) Fused in Sarcoma (FUS), (**C**) heterogeneous nuclear ribonucleoprotein A1 (hnRNPA1).

**A)**

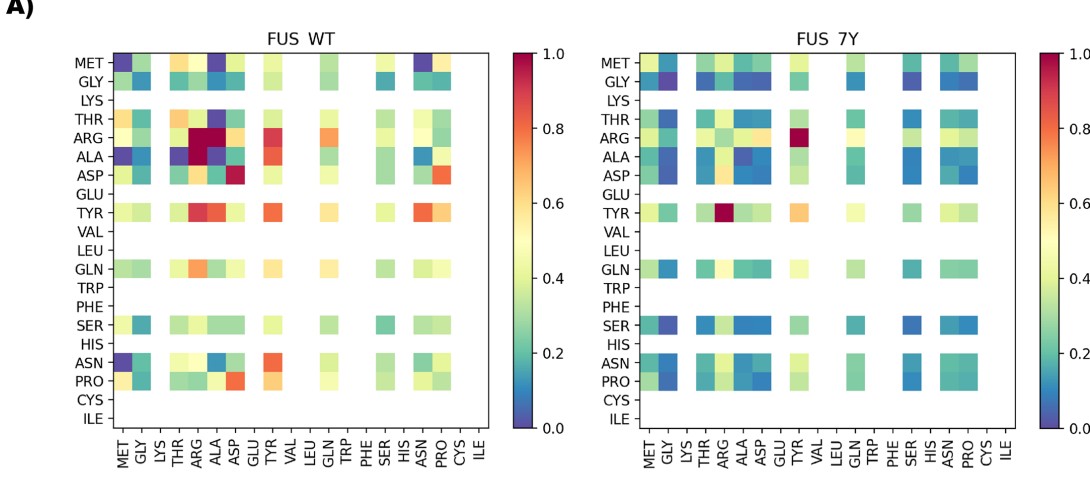

**B)**

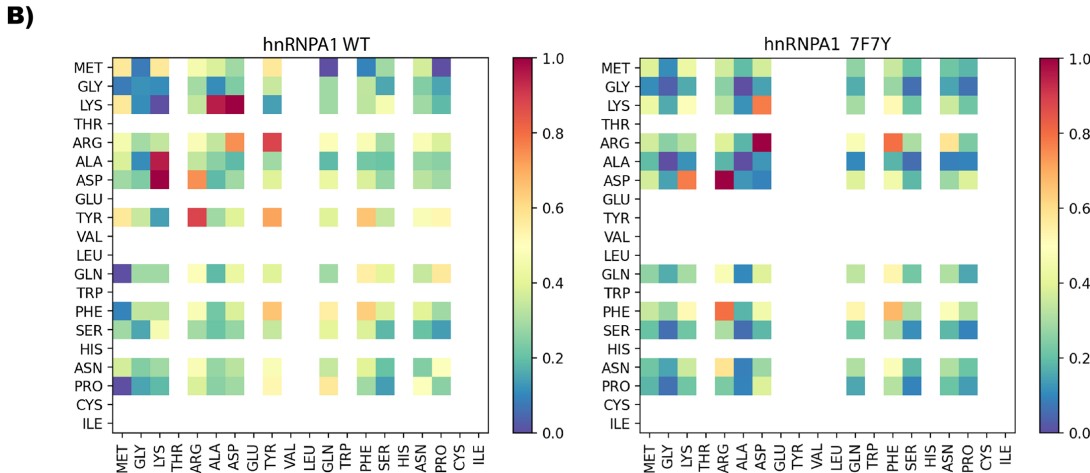

**Appendix 1—figure 10.** Normalized contact maps of different aromatic variants of prion-like low complexity domains (PLDs): (**A**) Fused in Sarcoma (FUS) (**B**) heterogeneous nuclear ribonucleoprotein A1 (hnRNPA1).

