## [Editor Report · eLife Assessment]

The authors performed extensive coarse-grained molecular dynamics simulations of 140 different prion-like domain variants to interrogate how specific amino acid substitutions determine the driving forces for phase separation. The analyses are **solid**, and the derived predictive scaling laws can aid in identifying potential phase-separating regions in uncharacterized proteins. Overall, this is a **valuable** contribution to the field of biomolecular condensates. It exemplifies how data-driven methodologies can uncover new insights into complex biological phenomena.

---

## [Referee Report · Reviewer #1 (Public review)]

Summary:

In this preprint, the authors systematically and rigorously investigate how specific classes of residue mutations alter the critical temperature as a proxy for the driving forces for phase separation. The work is well executed, the manuscript well-written, and the results reasonable and insightful.

Strengths:

The introductory material does an excellent job of being precise in language and ideas while summarizing the state of the art. The simulation design, execution, and analysis are exceptional and set the standard for large-scale simulation studies. The results, interpretations, and Discussion are largely nuanced, clear, and well-motivated, and the pedagogical nature with which sampling convergence is discussed is greatly appreciated. Finally, the underlying data are shared in a clear and accessible manner. Overall, the manuscript is a model

Weaknesses:

The simplicity of a one-bead-per-residue model parameterized to capture UCST-type phase behavior does perhaps impact some aspects of the generality of this work. That said, the authors carefully acknowledge these limitations, and overall, this is not seen as a major weakness of the conclusions drawn or the manuscript, given those conclusions are appropriately couched.

---

## [Referee Report · Reviewer #2 (Public review)]

This is an interesting manuscript where a CA-only CG model (Mpipi) was used to examine the critical temperature (Tc) of phase separation of a set of 140 variants of prion-like low complexity domains (PLDs). The key result is that Tc of these PLDs seems to have a linear dependence on substitutions of various sticker and space residues. This is potentially useful for estimating the Tc shift when making novel mutations of a PLD.

Comments on revisions: The authors have addressed concerns raised previously.

---

## [Referee Report · Reviewer #3 (Public review)]

Summary:

"Decoding Phase Separation of Prion-Like Domains through Data-Driven Scaling Laws" by Maristany et al. offers a significant contribution to the understanding of phase separation in prion-like domains (PLDs). The study investigates the phase separation behavior of PLDs, which are intrinsically disordered regions within proteins that have the propensity to undergo liquid-liquid phase separation (LLPS). This phenomenon is crucial in forming biomolecular condensates, which play essential roles in cellular organization and function. The authors employ a data-driven approach to establish predictive scaling laws that describe the phase behavior of these domains.

Strengths:

The study benefits from a robust dataset encompassing a wide range of PLDs, which enhances the generalizability of the findings. The authors' meticulous curation and analysis of this data add to the study's robustness. The scaling laws derived from the data provide predictive insights into the phase behavior of PLDs, which can be useful in the future for the design of synthetic biomolecular condensates.

---

## [Author Response]

The following is the authors’ response to the original reviews.

**Reviewer # 1 (Public Review):**
Summary:Inthispreprint, theauthorssystematicallyandrigorouslyinvestigatehowspecificclassesofresiduemutations alter the critical temperature as a proxy for the driving forces for phase separation. The work is well executed, the manuscript well-written, and the results reasonable and insightful.Strengths:The introductory material does an excellent job of being precise in language and ideas while summarizing the state of the art. The simulation design, execution, and analysis are exceptional and set the standard for these types of large-scale simulation studies. The results, interpretations, and Discussion are largely nuanced, clear, and well-motivated.

We thank the reviewer for their assessment of our work and for highlighting the key strengths of the paper.

Weaknesses:This is not exactly a weakness, but I think it would future-proof the authors’ conclusions to clarify a few key caveats associated with this work. Most notably, given the underlying implementation of the Mpipi model, temperature dependencies for intermolecular interactions driven by solvent effects (e.g., hydrophobic effect and charge-mediated interactions facilitated by desolvation penalties) are not captured. This itself is not a “weakness” per se, but it means I would imagine CERTAIN types of features would not be wellcaptured; notably, my expectation is that at higher temperatures, proline-rich sequences drive intermolecular interactions, but at lower temperatures, they do not. This is likely also true for the aliphatic residues, although these are found less frequently in IDRs. As such, it may be worth the authors explicitly discussing.

We also thank the reviewer for pointing out that a more detailed discussion of the model limitations is needed. The original Mpipi model was designed to probe UCST-type transitions (that are associative in nature) of disordered sequences. The reviewer is correct, that in its current form, the model does not capture LCST-type transitions that depend on changes in solvation of hydrophobic residues with temperature. We have amended the discussion to highlight this fact.

Similarly, prior work has established the importance of an alpha-helical region in TDP-43, as well as the role of aliphatic residues in driving TDP-43’s assembly (see Schmidt et al 2019). I recognize the authors have focussed here on a specific set of mutations, so it may be worth (in the Discussion) mentioning [1] what impact, if any, they expect transient or persistent secondary structure to have on their conclusions and [2] how they expect aliphatic residues to contribute. These can and probably should be speculative as opposed to definitive.Again - these are not raised as weaknesses in terms of this work, but the fact they are not discussed is a minor weakness, and the preprint’s use and impact would be improved on such a discussion.

We agree with the reviewer that the effects of structural changes/propensities on these scaling behaviors would be an interesting and important angle to probe. We also comment on this in the discussion.

**Reviewer # 2 (Public Review):**
This is an interesting manuscript where a CA-only CG model (Mpipi) was used to examine the critical temperature (Tc) of phase separation of a set of 140 variants of prion-like low complexity domains (PLDs). The key result is that Tc of these PLDs seems to have a linear dependence on substitutions of various sticker and space residues. This is potentially useful for estimating the Tc shift when making novel mutations of a PLD. However, I have strong reservations about the significance of this observation as well as some aspects of the technical detail and writing of the manuscript.

We thank the reviewer for their thoughtful and detailed feedback on the manuscript.

(1) Writing of the manuscript: The manuscript can be significantly shortened with more concise discussions. The current text reads as very wordy in places. It even appears that the authors may be trying a bit too hard to make a big deal out of the observed linear dependence.The manuscript needs to be toned done to minimize self-promotion throughout the text. Some of the glaring examples include the wording “unprecedented”, “our research marks a significant milestone in the field of computational studies of protein phase behavior ..”, “Our work explores a new framework to describe, quantitatively, the phase behavior ...”, and others.

We thank the reviewer for their suggestions on the writing of the manuscript. We understand the concern regarding the length and tone of the manuscript, and in response to their feedback, we have revised the language throughout the manuscript.

There is really little need to emphasize the need to manage a large number of simulations for all 140 variants. Yes, some thoughts need to go into designing and managing the jobs and organizing the data, but it is pretty standard in computational studies. For example, large-scale protein ligand-free energy calculations can require one to a few orders of magnitude larger number of runs, and it is pretty routine.

We fully agree with the reviewer that this aspect of the study is relatively standard in computational research and does not require special emphasis. In response, we have revised the manuscript to shorten the aforementioned section, focusing instead on the scientific insights gained from the simulations rather than the logistical challenges of managing them.

When discussing the agreement with experimental results on Tm, it should be noted that the values of R > 0.93 and RMSD < 14 K are based on only 16 data points. I am not sure that one should refer to this as “extended validation”. It is more like a limited validation given the small data size.

We thank the reviewer for their consideration of our validation set. Indeed, the agreement with experimental results is based on 16 data points, as this set represents the available published data at the time of writing of this manuscript. The term “extended validation” is used to signify that our current dataset builds upon previous validations (in Joseph, Reinhardt et al. Nat Comput. Sci. 2021), incorporating additional variants not previously examined. The metrics of an r>0.93 and a low RMSD indicate a strong agreement between the model and experiments, and an improvement with respect to other reported models. We are committed to continue validating our methods.

Results of linear fitting shown in Eq 4-12 should be summarized in a single table instead of scattering across multiple pages.

We considered the reviewer’s suggestion to compile all the laws into a single table. However, we believe it would be more effective for readers to reference each relationship directly where it is first discussed in the text. That said, we do include Table 1 in the original manuscript, which provides a summary of all the laws.

The title may also be toned down a bit given the limited significance of the observed linear dependence.

We respectfully disagree with the reviewer and believe that the current title accurately captures the scope of the manuscript.

(2) Significance and reliability of Tc: Given the simplicity of Mpipi (a CA-only model that can only describe polymerchaindimension)andthelowcomplexitynatureofPLDs, thesequencecompositionitselfisexpected to be the key determinant of Tc. This is also reflected in various mean-field theories. It is well known that other factors will contribute, such as patterning (examined in this work as well), residual structures, and conformational preferences in dilute and dense phases. The observed roughly linear dependence is a nice confirmation but really unsurprising by itself. It appears how many of the constructs deviate from the expected linear dependence (e.g., Figure 4A) may be more interesting to explore.

While linear dependencies in critical solution temperatures may appear expected for certain systems, for example, symmetric hard spheres, the heterogeneity of intrinsically disordered regions (IDRs), like prion-like domains (PLDs), make this finding notable. The simplicity of our linear scaling law belies the underlying complexity of multivalent interactions and sequence-dependent behaviors in a certain sequence regime, which has not been quantitatively characterized in this manner before. Likewise, although linear dependencies may be expected in simplified models, the real-world applicability and empirical validation of these laws in biologically relevant systems are not guaranteed. Our chemically based model provides the robustness needed to do that. The linear relationship observed is significant because it provides a predictive framework for understanding how specific mutations affect a diverse set of PLDs. The framework presented can be extended to other protein families upon the application of a validated model, which might or might not yield linear relationships depending on the cooperative effects of their collective behavior. This extends beyond confirming known theories—it offers a practical tool for predicting phase behavior based on sequence composition

We agree with the reviewer that, while the overarching linear trend is clear, deviations from linearity observed in constructs like those in Figure 4A point to additional, and interesting, layers of complexity. These deviations offer interesting avenues for future research and suggest that while linearity might dominate PLD critical behavior, other factors may modulate this behavior under specific conditions.

This is an excellent suggestion from the reviewer that, while it falls outside the scope of the current study, we are interested in exploring in the future.

Finally, the relationships are all linear, they have been normalized in different ways—the strength of the study also lies in that. Instead of focusing solely on linearity, our study explores the physical mechanisms that underlie these relationships. This approach provides a more complete understanding of how sequence composition and the underlying chemistry of the mutated residues influence *T*_c</sub._

The assumption that all systems investigated here belong to the same universality class as a 3D Ising model and the use of Eqn 20 and 21 to derive Tc is poorly justified. Several papers have discussed this issue, e.g., see Pappu Chem Rev 2023 and others. Muthukumar and coworkers further showed that the scaling of the relevant order parameters, including the conserved order parameter, does not follow the 3D Ising model. More appropriate theoretical models including various mean field theories can be used to derive binodal from their data, such as using Rohit Pappu’s FIREBALL toolset. Imposing the physics of the 3D Ising model as done in the current work creates challenges for equivalence relationships that are likely unjustified.

We thank the reviewer for raising this point and for highlighting the FIREBALL toolset. Based on our understanding, FIREBALL is designed to fit phase diagrams using mean-field theories, such as Flory–Huggins and Gaussian Cluster Theory. Our experience with this toolset suggests that it places a higher weight on the dilute arm of the binodal. However, in our slab simulations, we observe greater uncertainty in the density of the dilute arm. This leads to only a moderate fit of the data to the mean-field theories employed in the toolset. While we agree that there is no reason to assume the phase behavior of these systems is fully captured by the 3D Ising model, we expect that such a model will describe the behavior near the critical point better than mean-field theories. Testing our results further with different critical exponents would be valuable in assessing how these predictions compare to a broader set of experimental data. Additionally, we have made the raw data points for the phase diagrams available on our GitHub, enabling practitioners to apply alternative fitting methods.

While it has been a common practice to extract Tc when fitting the coexistence densities, it is not a parameter that is directly relevant physiologically. Instead, Csat would be much more relevant to think about if phase separation could occur in cells.

WhileitistruethatCsatisdirectlyrelevanttowhetherphaseseparationcanoccurincellsunder physiological conditions, *T*_c_ should not be dismissed as irrelevant.*T*_c_ provides fundamental insights into the thermodynamics of phase separation, reflecting the overall stability and strength of interactions driving condensate formation. This stability is crucial for understanding how environmental factors, such as temperature or mutations, might affect phase behavior. In Figure 2C and D we compare experimental *C*_sat_ values with our predicted *T*_c_ from simulations. These quantities are roughly inversely proportional to each other and so we expect that, to a first approximation, the relationships recovered for *T*_c_ should hold when considering*C*_sat_ at a fixed temperature.

**Reviewer # 3 (Public Review):**
Summary:“Decoding Phase Separation of Prion-Like Domains through Data-Driven Scaling Laws” by Maristany et al. offers a significant contribution to the understanding of phase separation in prion-like domains (PLDs). The study investigates the phase separation behavior of PLDs, which are intrinsically disordered regions within proteins that have a propensity to undergo liquid-liquid phase separation (LLPS). This phenomenon is crucial in forming biomolecular condensates, which play essential roles in cellular organization and function. The authors employ a data-driven approach to establish predictive scaling laws that describe the phase behavior of these domains.Strengths:The study benefits from a robust dataset encompassing a wide range of PLDs, which enhances the generalizability of the findings. The authors’ meticulous curation and analysis of this data add to the study’s robustness. The scaling laws derived from the data provide predictive insights into the phase behavior of PLDs, which can be useful in the future for the design of synthetic biomolecular condensates.

We thank the reviewer for highlighting the importance of our work and for their critical feedback.

Weaknesses:While the data-driven approach is powerful, the study could benefit from more experimental validation. Experimental studies confirming the predictions of the scaling laws would strengthen the conclusions. For example, in Figure 1, the Tc of TDP-43 is below 300 K even though it can undergo LLPS under standard conditions. Figure 2 clearly highlights the quantitative accuracy of the model for hnRNPA1 PLD mutants, but its applicability to other systems such as TDP-43, FUS, TIA1, EWSR1, etc., may be questionable.

In the manuscript, we have leveraged existing experimental data for the A1-LCD variants, extracting critical temperatures and saturation concentrations to compare with our model and scaling law predictions. We acknowledge that a larger set of experiments would be beneficial. By selecting sequences that are related, we hypothesize that the scaling laws described herein should remain robust. In the case of TDP-43, to our knowledge this protein does not phase separate on its own under standard conditions. In vitro experiments that report phase separation at/above 300 K involve either the use of crowding agents (such as dextran or PEG) or multicomponent mixtures that include RNA or other proteins. Therefore, our predictions for TDP-43 are consistent with experiments. In general, we hope that the scaling laws presented in our work will inspire other researchers to further test their validity.

The authors may wish to consider checking if the scaling behavior is only observed for Tc or if other experimentally relevant quantities such as Csat also show similar behavior. Additionally, providing more intuitive explanations could make the findings more broadly accessible.

In Figure 2C and D we compare experimental *C*_sat_ values with our predicted *T*_c_ from simulations. These quantities are roughly inversely proportional to each other and so we expect that, to a first approximation, the relationships recovered for *T*_c_ should hold when considering *C*_sat_ at a fixed temperature.

The study focuses on a particular subset of intrinsically disordered regions. While this is necessary for depth, it may limit the applicability of the findings to other types of phase-separating biomolecules. The authors may wish to discuss why this is not a concern. Some statements in the paper may require careful evaluation for general applicability, and I encourage the authors to exercise caution while making general conclusions. For example, “Therefore, our results reveal that it is almost twice more destabilizing to mutate Arg to Lys than to replace Arg with any uncharged, non-aromatic amino acid...” This may not be true if the protein has a lot of negative charges.

A significant number of proteins, in addition to those mentioned in the manuscript, that contain prion-like low complexity domains have been reported to exhibit phase separation behaviors and/or are constituents of condensates inside cells. We therefore expect these laws to be applicable to such systems and have further revised the text to emphasize this point. As the reviewer suggests, we have also clarified that the reported scaling of various mutations applies to these systems.

I am surprised that a quarter of a million CPU hours are described as staggering in terms of computational requirements.

We have removed the note on CPU hours from the manuscript. However, we would like to clarify that the amount of CPU hours was incorrectly reported. The correct estimate is 1.25 million hours, but this value was unfortunately misrepresented during the editing process. We thank the reviewer for catching this mistake on our part.

**Reviewer # 1 (Recommendations For The Authors):**
Some minor points here:“illustrating that IDPs indeed behave like a polymer in a good solvent [43]. ” Whether or not an IDP depends as a polymer in a good solvent depends on the amino acid sequence - the referenced paper selected a set of sequences that do indeed appear on average to map to a good-solvent-like polymer, but lest we forget SAXS experiments require high protein concentrations and until the recent advent of SEC-SAXS, your protein essentially needed to be near infinitely soluble to be measured. As such, this paper’s conclusions are, apparently, ignorant of the limitations associated with the data they are describing, drawing sweeping generalizations that are clearly not supported by a multitude of studies in which sequence-dependencies have led to ensembles with a scaling exponent far below 0.59 (See Riback et al 2017, Peng et al 2019, Martin et al 2020, etc).

We thank the reviewer for raising this point. To avoid making incorrect generalizations and potentially misleading readers, we have removed the quoted statement from our manuscript.

As of right now, the sequences are provided in a convenient multiple-sequence alignment figure. However, it would be important also to provide all sequences in an Excel table to make it easy for folks to compare.

In addition to the sequence alignment figure, we now provide all tested sequences in an Excel table format in the GitHub repository.

Maybe I’m missing it, but it would be extremely valuable if the coexistence points plot in all the figures were provided as so-called source data; this could just be on the GitHub repository, but I’m envisaging a scenario where for each sequence you have a 4 column file where Col1=concentration and Col2=temperature, col3=fit concentration and col4=fit temperature, such that someone could plot col1 vs. col2 and col3 vs. col4 and reproduce the binodals in the various figures. Given the tremendous amount of work done to achieve binodals:

The coexistence points used to plot the figures are now provided in the GitHub, in a format similar to that suggested by the reviewer.

It would be nice to visually show how finite size effects are considered/tested for (which they are very nicely) because I think this is something the simulation field should be thinking about more than they are.

Thank you for highlighting this point. In our previous work (supporting information of the original Mpipi paper), we demonstrated a thorough approach by varying both the cross-sectional area of the box and the long axis while keeping the overall density constant. In this work, we verified that the cross-sectional area was larger than the average *R*_g_ of the protein. We then maintained a fixed cross-sectional area to long-axis ratio, varying the number of proteins while keeping the overall density constant. We have updated Appendix 1–Figure 2 to clarify our procedure and revised the caption to better explain how we ensured the number of proteins was adequate.

When explaining the law of reticular diameters, it would be good to explain where the 3.06 exponent comes from.

Based on the reviewer’s suggestion, we have added to the text: “The constant 3.06 in the equation is a dimensionless empirical factor that was derived from simulations of the 3D Ising model.”

The NCPR scale in Figure 5 being viridis is not super intuitive and may benefit from being seismic or some other r-w-b colormap just to make it easier for a reader to map the color to meaning.

We thank the reviewer for this suggestion and have replaced the scale with a r-w-b colormap.

The “sticker and spacer” framework has received critiques recently given its perceived simplicity. However, this work seems to clearly illustrate that certain types of residues have a large effect on Tc when mutated, whereas others have a smaller effect. It may be worth re-phrasing the sticker-spacer introduction not as “everyone knows aromatic/arginine residues are stickers” but as “aromatic and arginine residues have been proposed to be stickers, yet other groups have argued all residues matter equally” and then go on to make the point that while a black-and-white delineation is probably not appropriate, based on the data, certain residues ARE demonstrably more impactful on Tc than others, which is the definition of stickers. With this in mind, it may be useful to separate out a sticker and a spacer distribution in Figure 1D, because the different distribution between the two residues types is not particularly obvious from the overlapping points.

We have revised the introduction of the sticker–spacer model in the manuscript for clarity. As the reviewer suggests, we have also separated the sticker and spacer distribution, which is now summarized in new Appendix 0–figure 8.

**Reviewer # 3 (Recommendations For The Authors):**
Figure 2 clearly highlights the quantitative accuracy of the model for hnRNPA1 PLD mutants, but its applicability to other systems such as TDP-43, FUS, TIA1, EWSR1, etc., may be questionable. The following sentence may be revised to reflect this: “Our extended validation set confirms that the Mpipi potential can ...”

Based on the reviewer’s suggestion, we have revised the text: “Our validation set, which expands the range of proteins variants originally tested [32], highlights that the Mpipi potential can effectively capture the thermodynamic behavior of a wide range of hnRNPA1-PLD variants, and suggests that Mpipi is adequate for proteins with similar sequence compositions, as in the set of proteins analyzed in this study. In recent work by others [66], Mpipi was tested against experimental radius of gyration data for 137 disordered proteins and the model produced highly accurate results, which further suggests the applicability of the approach to a broad range of sequences.”